

# Density-operator evolution:
# Complete positivity and the Keldysh real-time expansion

Viktor Reimer[1] and Maarten Rolf Wegewijs[1,2,3⋆]

**1** Institute for Theory of Statistical Physics, RWTH Aachen, 52056 Aachen, Germany
**2** Peter Grünberg Institut, Forschungszentrum Jülich, 52425 Jülich, Germany
**3** JARA-FIT, 52056 Aachen, Germany

⋆ m.r.wegewijs@fz-juelich.de

## Abstract

We study the reduced time-evolution of general open quantum systems by combining insights from quantum-information and statistical field theory. Inspired by prior work [Eur. Phys. Lett. **102**, 60001 (2013) and Phys. Rev. Lett. **111**, 050402 (2013)] we establish the explicit structure guaranteeing the complete positivity (CP) and trace-preservation (TP) of the real-time evolution expansion in terms of the microscopic system-environment coupling.

This reveals a fundamental *two-stage* structure of the coupling expansion: Whereas the first stage naturally defines the dissipative timescales of the system –*before* having integrated out the environment completely– the second stage sums up elementary physical processes, each described by a CP superoperator. This allows us to establish the highly nontrivial functional relation between the (Nakajima-Zwanzig) memory-kernel *superoperator* for the reduced density operator and novel memory-kernel *operators* that generate the Kraus operators of an operator-sum. We illustrate the physically different roles of the two emerging coupling-expansion parameters for a simple solvable model. Importantly, this operational approach can be implemented in the existing Keldysh real-time technique and allows approximations for *general time-nonlocal* quantum master equations to be systematically compared and developed while keeping the CP and TP structure explicit.

Our considerations build on the result that a Kraus operator for a physical measurement process on the environment can be obtained by 'cutting' a group of Keldysh real-time diagrams 'in half'. This naturally leads to Kraus operators *lifted* to the system *plus environment* which have a diagrammatic expansion in terms of time-nonlocal memory-kernel operators. These lifted Kraus operators obey coupled time-evolution equations which constitute an unraveling of the original Schrödinger equation for system plus environment. Whereas both equations lead to the same reduced dynamics, only the former explicitly encodes the operator-sum structure of the coupling expansion.

# 1   Introduction

The experimental progress in nanoscale and mesoscopic devices has continued to drive the development of theoretical methods to tackle models of nonequilibrium quantum systems that interact with their environment [1]. Open-system approaches based on the reduced density operator are particularly suitable for dealing with the strong local interaction effects that are often central to device operation, for example, in read-out circuits of qubits and many quantum transport devices. These interactions together with the *continuum* of environment energies often complicate the analysis of the dynamics when the system-environment coupling becomes strong.

In some limits, the evolution of the density operator is known to exhibit a special Markovian semi-group property. For this case Gorini, Kossakowski, Sudarshan [2] and Lindblad [3] (GKSL) have established a form of the underlying quantum master equations that is both necessary and sufficient for the dynamics to be completely positive (CP) and trace-preserving (TP), cf. also Ref. [4]. Thus, in these specific limits only quantum master equations of this form guarantee that the evolved reduced density operator still makes statistical sense which is one of the reasons why GKSL master equations have become popular and successful. More recently, such easily verified restrictions on the form of quantum master equations have been extended, see, e.g., Refs. [5–8].

However, the experimental importance of effects due to strong coupling and non-Markovianity– which are tied together [9]– has spurred progress in a variety of other approaches: Inclusion of parts of the environment into the system [10], time-convolutionless master equations [11, 12], stochastic descriptions such as quantum trajectories [13], path-integral [14], quantum Monte Carlo (QMC) [15], and hierarchical methods [16, 17], multi-layer multiconfiguration time-dependent Hartree method [18], perturbative expansions [19–25], resummation [26–28] and related techniques [29–31], projection techniques [32, 33] and real-time renormalization-group methods [34–41]. These are applicable to more general reduced dynamics derived from unitary evolution $U(t)$ of initially uncorrelated states $\rho(0)$ of the system (S) and $\rho^{\mathrm{E}}$ of the environment (E):

$$\rho(t) := \Pi(t)\rho(0) = \underset{\mathrm{E}}{\mathrm{Tr}}\, U(t)\big[\rho(0) \otimes \rho^{\mathrm{E}}\big]U^{\dagger}(t). \tag{1}$$

We use units $\hbar = k_{\mathrm{B}} = |e| = 1$ and –unless stated otherwise– work in the interaction picture with respect to the system-environment coupling $V(t)$ which generates the unitary evolution $U(t)$. The dynamics (1) is always CP and TP [42] but need not have the GKSL form. In this general setting it is unavoidable to make approximations at some stage. Although these aim to approach the exact result as close as possible, a key challenge is to ensure that they do not uproot the CP and TP properties of the reduced dynamics. Simultaneously one wants to maintain a clear view on microscopic contributions to physical processes. Meeting these *different* goals is a particularly difficult task.

## 1.1   Complementary approaches to reduced dynamics

In this paper we make a further [43, 44] step towards clarifying this challenging issue in a general setting by combining complementary insights from quantum information theory and statistical physics. On the one hand, we are guided by the *operator-sum form* of the general reduced density operator dynamics,

$$\rho(t) = \sum_{e} K^{e}(t)\rho(0)K^{e\dagger}(t), \tag{2}$$

developed by Sudarshan, Mathews, and Rau [45] and Kraus [46], cf. Ref. [47]. Dynamics of the type (1) can always be written in this form. The quadratic form manifests the CP

property whereas TP is enforced by the sum rule $\sum_e K^{e\dagger}(t)K^e(t) = \mathbb{1}_S$. The sum (2) averages the evolution over all outcomes $e$ of *possible measurements* on the inaccessible environment. Although it is well-known how the Kraus operators $K^e(t)$ –acting only on the system– can be expressed in the joint unitary $U(t)$ and the initial environment state $\rho^E$, the direct evaluation of these quantities becomes intractable when dealing with microscopic models with a *continuous* environment and strong local interactions. Obtaining an explicit Kraus form in such settings is usually restricted to simple models which can either be solved exactly or treated in the simplifying GKSL limit. Besides guaranteeing the CP property, access to the individual Kraus operators also allows the calculation of additional measures of information yielding interesting insights, even for well-known solvable models [48].

Approaches to approximately solve a broader range of models have been developed in statistical physics where one instead considers general kinetic equations of the form

$$\frac{d}{dt}\rho(t) = \int_0^t dt' \Sigma(t,t')\rho(t') \equiv [\Sigma * \rho](t). \tag{3}$$

The time-nonlocal nature of the self-energy or memory kernel superoperator $\Sigma(t,t')$ and the time convolution $*$ in Eq. (3) underscore the non-Markovian nature of the general dynamics (1). The kinetic equation (3) is derived equivalently [11,23] either by Nakajima-Zwanzig projection superoperators or by the Keldysh real-time expansion[1] on which we will focus here. In either way what makes the practical evaluation feasible is the exploitation of *Wick factorization* of environment correlations in the cases of practical interest where the environment is noninteracting and initially in a mixed thermal state. This allows one to account for higher-order contributions in the system-environment coupling and renormalization effects which may be crucial for the phenomena of interest. Noting that Wick's theorem can be generalized (cf. Ref. [49] and App. A), we stress that the kinetic method is in general on equal footing with the operator-sum approach.

It must thus be possible to express the dynamical map $\rho(t) = \Pi(t)\rho(0)$ that solves Eq. (3) in the form of an operator sum, $\Pi(t) = \sum_e K^e(t) \bullet K^{e\dagger}(t)$, letting $\bullet$ indicate the position of the argument. How to actually do this on the level of the microscopic coupling $V(t)$ has remained unclear until recently. In this paper we will find the explicit nontrivial relation between the two fundamental equations (2) and (3), in particular between $\Sigma(t,t')$ and $K^e(t)$, at every step linking the two complementary approaches. Following up earlier work in this direction [43,44] we develop a transparent set of diagrammatic rules by which all of this can be achieved. This allows for a *microscopic* understanding of the CP-TP structure of the reduced evolution $\Pi(t)$ and its memory kernel $\Sigma(t,t')$ in terms of quantities that are relevant in practical calculations for open systems with continuous environments. This is a crucial prerequisite for an improved understanding of approximations.

## 1.2 Dynamics and entanglement

The key idea of our work is to set up the Wick expansion in terms of physical, i.e., *operational* quantities. These are quantities which can be explicitly expressed in terms of quantum postulates (preparation, evolution, measurement) and can thus be implemented as quantum circuits. We are motivated by the fact that other technical developments in many-body approaches such as DMRG [50–52] and tensor networks [53–55] have profited from quantum information theory precisely by its insistence on physical / operational clarity, in particular regarding entanglement. The CP property that we focus on here concerns the importance of entanglement for dynamical maps. This is well-understood from two opposing angles: on the

---

[1]This technique applied to density operators is similar but clearly distinct from the technique for non-equilibrium *Green's functions* which also goes by the name of Keldysh.

one hand, the *requirement of CP* due to entanglement in the theory of evolution [56], and on the other the *failure of CP* for positivity-preserving (PP) maps used to 'witness' or detect entanglement [57].

Despite this, the role of entanglement seems to have been hardly explored within field-theoretical approaches to open-system evolution. Although a great deal of useful intuition and technical expertise for Keldysh time-evolution diagrams has been developed, their precise operational meaning in terms of physical measurements has remained unclear. The present work fills this gap by identifying which sums of standard Keldysh diagrams are physically / operationally meaningful. To this end we relate the perturbation expansion to a quantum circuit in which evolution is conditioned on *partial* measurements of the open system's environment. To achieve this, it is necessary to purify both the state of the environment as well as the unitary evolution map $U(t)$. By insisting on operational quantities we thus automatically obtain a new *diagrammatic* expansion of (operators underlying) the *Kraus* operators in terms of the microscopic system-environment coupling. The converse implication is that one can exploit field-theoretical methods in the *general* setting of quantum information theory. Although our analysis focuses on the density-operator method, it relates to similar developments in non-equilibrium Green's functions [58] and quantum-field theory [59] (Cutkosky 'cutting rules').

As mentioned, our work was stimulated in particular by two prior publications [43, 44] and aims to complement these, cf. also Refs. [60–63]. Our formalism extends and generalizes the scope of the important Refs. [43, 62, 63] whose formulation is most easily connected to the standard density-operator Keldysh formalism [26]. In particular, we allow for multilinear coupling $V$ to both fermionic and bosonic environments and provide rules for switching back and forth between the Kraus description (2) and the quantum kinetic description (3). Ref. [44], on the other hand, ingeniously circumvents positivity problems by using a projection technique to derive the evolution of $\sqrt{\rho(t)}$ and squaring the result to obtain a positive density operator. While both approaches at first seem quite distinct, they turn out to be closely related. This is nontrivial to see directly, but becomes clear once they are tied to the basic operational formulation of quantum information theory. Whereas the present paper achieves this operational formulation for the Keldysh diagrammatic real-time approach related to Refs. [43, 62, 63], the projection-superoperator formulation of Ref. [44] will be addressed in subsequent work [64]. This underscores the extended scope[2] of the ideas developed here. Throughout the paper we will point out the extensions and advantages of our approach relative to these works.

## 1.3 Microscopic models for open-system dynamics

The open-system models we are interested in generate dissipative dynamics by coupling a strongly interacting discrete quantum system to mixed-state reservoirs with *continuous* spectra. We allow for spin and orbital degrees of freedom and multiple reservoirs at independent temperatures and electrochemical potentials (electro-thermal bias). Also, the coupling to these *continuous* reservoirs can depend on spin and orbital indices (noncollinearity effects) and on frequency ($\omega$) for different reservoir bands of finite width. Moreover, the system and environment modes can be either bosonic or fermionic or some hybrid. However, what makes these models of interest difficult to treat is that we allow for strong local interactions. These really necessitate the consideration of approximations and motivates the combination of quantum information and statistical field theory.

To establish contact between these complementary approaches, the crucial quantities are the *environment modes* which we will denote by $m$. We will see that in the statistical field ap-

---

[2]The projection approach [33,56] based on the quantum superchannel formalism is more general by accounting for an initially correlated system-environment state. It is an interesting open question how this can be related to the present approach and that of Refs. [43, 44, 62, 63].

proach, these modes label the contraction lines of Keldysh diagrams, whereas in the quantum information approach they play the role of measurement outcomes labeling Kraus operators. The environment modes are most easily introduced by way of an example. Consider a fermion mode coupled to a continuous reservoir of noninteracting fermions at temperature $T$ and electrochemical potential $\mu$, a model that we revisit in Sec. 5 in the $T \to \infty$ limit. The total Hamiltonian of this model generating the unitary evolution $U(t) = e^{-iH^{\text{tot}}t}$ in Eq. (1) is given by

$$H^{\text{tot}} = H^0 + V = \varepsilon\, d^\dagger d + \int_\omega \omega\, b_\omega^\dagger b_\omega + \sqrt{\frac{\Gamma}{2\pi}} \int_\omega \left( d^\dagger b_\omega + b_\omega^\dagger d \right). \tag{4}$$

The system (field $d$) is bilinearly coupled to the modes in the environment (field $b_\omega$) by tunnel rates $\Gamma$ that are independent of their energy $\omega$ (wideband limit), denoting $\int_\omega := \int_{-\infty}^{\infty} d\omega$. Letting a particle-hole index $\eta$ indicate a creation ($\eta = +$) or annihilation ($\eta = -$) operator[3] the coupling can be written as

$$V = -\sum_{\eta_1} \int_{\omega_1} \sqrt{\frac{\Gamma}{2\pi}}\, \eta_1 d_{\eta_1}^\dagger b_{\eta_1 \omega_1}. \tag{5}$$

Thus, in this example we label one *mode of the environment* by a discrete particle-hole index and a continuous energy over which we sum and integrate, respectively:

$$m_1 = \eta_1 \omega_1. \tag{6}$$

An advantage of the diagrammatic approach of statistical field theory is that it provides a uniform treatment of all models of interest: The primary quantities of importance are the environment modes (6) with which the system has interacted. This is precisely what is needed to connect to quantum information theory where the operational formulation of the CP-TP dynamics hides all details except for the outcomes of measurements performed on the environment.

Thus, when one is interested in the physics of transport through quantum dot systems, one may extend the above simple model by including spin $\sigma = \uparrow, \downarrow$ on the electron fields $d \to d_\sigma$ and $b_\omega \to b_{\omega\sigma}$ and add a quartic interaction term $U d_\uparrow^\dagger d_\uparrow d_\downarrow^\dagger d_\downarrow$. In this case, the environment modes $m_i = \eta_i \omega_i \sigma_i$ include the reservoir spin. The so obtained Anderson impurity model exhibits nontrivial many-body physics such as the Kondo effect. If one is instead interested in an effective model focusing on Kondo scattering [65, 66], one has to deal with a coupling $V$ which is *bi-quadratic* in the system and environment fields, i.e., $\propto d_\sigma^\dagger d_{\sigma'}$ and $\propto b_{\omega\sigma}^\dagger b_{\omega'\sigma'}$. Although this alters the details of the diagrams (double vertices), the environment modes are the same. Similar considerations can be made for strongly interacting bosonic and hybrid models of interest in cold atoms, chemical dynamics, and quantum optics. We stress that *none* of the above model details play a role in the following: Only the environment modes will enter at a crucial point.

## 1.4 Approximations

Our motivation by approximations may seem to be at odds with quantum information theory which, by its insistence on operationally well-defined quantities, complicates rather than simplifies the problem. Indeed, if one strictly insists on dynamics that is *both* CP and TP, one is practically condemned to studying only a very limited set of open-system models and physical phenomena. Here we are emphatically *not* interested in such situations where one takes

---

[3]We use the convention that system and reservoir fields *anticommute,* requiring the fermion sign $\eta_1$.

the 'easy way out' by relying on one of several simplifications: (i) Models described by GKSL equations where CP and TP are encoded into the simple form of the generator. (ii) Models with exactly solvable non-GKSL dynamics where CP and TP can be seen explicitly [48]. (iii) Approximations that reduce[4] models to these cases.

We instead focus on the difficult problem of how general *many-body* approximations preserve the CP and TP properties of the dynamics. We follow the rationale –common in physics– that it may be better to disregard certain 'sum rules', then exploit this to set up approximations in terms of 'systematic' expansion parameters and afterwards check that the ignored 'sum rules' are fulfilled up to a 'small and controllable error'. How all these quoted terms are understood and implemented in detail is an important issue which, however, strongly depends on the problem at hand, the approach chosen and even the scientific community. What is of interest here is that independent of context there are two fundamentally different strategies for making approximation to open-system dynamics. We now outline these in a deliberately polarizing manner to emphasize their duality.

*TP approximations.* One approach is to enforce TP and check CP afterwards as a means of gauging the quality of the approximation. As it turns out, most approximations formulated in terms of the memory kernel $\Sigma(t, t')$ of the kinetic equation (3) have no problem with guaranteeing the trace-preservation but impose no restrictions at all to ensure complete positivity. Because TP is a linear constraint, it is less difficult to formulate very advanced approximation schemes which by their accuracy are likely to be CP. Nevertheless, for such TP approximations one cannot be sure in advance that (complete) positivity is not lost, implying that negative density operators may come out depending on the input. Actual loss of positivity of the state may be fatal for a calculation: It implies that the results cannot even be taken as a 'rough indication' of what is going on since negative probabilities are meaningless. A more subtle point that is often overlooked is that approximations may *still* fail even when they produce dynamics with positive outputs for all input states: Negative states may nevertheless appear when evolving part of an *entangled* composite quantum system. This is the loss of *complete* positivity of the dynamics, a property specific to quantum systems [56] which the exact evolution (1) of interest *also* possesses.

These issues have become more important due to the current interest from the side of statistical physics and quantum thermodynamics to formulate approximation schemes for *entropic* quantities. These are not well-defined unless the quantum state $\rho(t)$ is positive for all times. In this context the Keldysh approach has proven useful by its extension to 'parallel-worlds' [25] to compute Renyi entropies. Our operator-sum formulation of the Keldysh approach identifies a variety of approximations that give meaningful entropic quantities, including also the exchange entropy [67, 68] associated with the environment [48] which requires the evolution $\Pi(t)$ to be *completely* positive.

*CP approximations.* Clearly, a complementary approach is to enforce CP and check TP afterwards. Because of the nonlinearity of the CP constraint this is much harder and has not been explored much. Nevertheless, we will find it is not difficult to at least *characterize* large families of approximations which strictly guarantee complete positivity while offering large freedom to construct approximations of varying quality. The resulting CP maps are not TP but still trace-nonincreasing (TNI) and therefore make perfect sense as evolutions conditioned on a *subset* of measurement outcomes ('postselection') as used throughout quantum information theory. Nevertheless, failure of CP approximations to guarantee TP can be fatal in a different way: Although perhaps less important for short-time dissipative dynamics, TP is closely tied to the stationary state reached at large times. Also, the difficulty of evaluating CP-approximations may limit the accuracy so much as that more accurate TP-approximations may be a better way

---

[4]For example, CP-TP may become tractable by imposing a slave particle approximation. However, this may poorly capture full many-body phenomena such as the Kondo problem.

of achieving dynamics that turns out to be CP *a posteriori*.

Clearly, such CP approximations connect to the interests of quantum information theory where this property is essential. For example, evaluating Eq. (1) for the evolution $\Pi(t)$ using some non-CP approximation would break the chain of reasoning because fundamental insights in this field rely on this property, such as the quantum data-processing inequality [67]. In fact, the CP property has recently been shown to be *equivalent* to this most fundamental inequality of quantum information-processing [69]. Since the approach we develop here is centered around the *time-evolution of* (operators underlying the) *Kraus operators*, it provides a systematic way for going beyond the limited GKSL approach *without* giving up CP. Importantly, we show how this can be done by exploiting well-known statistical physics techniques. Finally, we note that the characterization of CP approximations is already interesting by itself since it provides a guide to TP approximations that is currently missing: It indicates how in TP-approximations one can systematically 'improve towards complete positivity' by identifying which diagrams should be added to give a *better partial* account of a physical process without insisting on strict CP.

*CP-TP duality.* Our work does not aim to decide on any general debate of CP vs. TP approximations. The merits and mutual points of critique of the above outlined approaches are all valid and in neither approach there is a general way of overcoming all challenges. The point we wish to emphasize is that – excluding the above mentioned 'easy ways out' – the trade-off between CP and TP approximations is in general nontrivial. What is perhaps most interesting is that the two approximation strategies and their problems are *fundamentally* tied together. It has been noticed [70] that there is a kind of duality between complete positivity and trace-preservation: In practice, rigorously fixing CP in some approximation tends to uproot TP and *vice versa*. Although this is not entirely unexpected based on the Choi-Jamiołkowski correspondence between quantum states and evolutions (cf. App. C), we will identify the *microscopic* origin of this duality in terms of elementary diagrammatic contributions to the real-time expansion. This shows that in approximation schemes based on a selection of contributions / diagrams, CP and TP cannot be achieved *simultaneously*. Recent exceptional approximations that are both CP *and* TP require more sophisticated schemes explored in Refs. [43, 44, 60–63] which we will also touch upon.

In the present paper we focus on the prerequisites for a clearer understanding of the above mentioned approximation schemes and the underlying *physical / operational reason* for the mutually exclusive nature of the difficulties with CP and TP. This is of common interest to both research fields: Although our encompassing formalism remains geared towards applications –by its formulation in terms of microscopic couplings and Keldysh diagrams of Eq. (3)– it is firmly rooted in the operator-sum (2) and defined operationally in terms of quantum circuits.

## 1.5 Outline and guide

The paper is organized as follows. Sec. 2 first applies methods of quantum information to the problem: We review the the importance of entanglement for dynamics as expressed by complete positivity and the operator-sum representation. This leads us to purify the environment state –expressing finite temperature in terms of entanglement– and also purify the *evolution* map using Wick normal-ordering. These physically / operationally motivated steps reveal clearly how measurements condition the time-evolution and connect the CP structure with Wick's theorem.

In Sec. 3 we turn to statistical field theory and connect the obtained operator-sum representation with Keldysh real-time diagrams. This allows us to identify *groups* of diagrams associated with a physical process conditioned on a measurement outcome. Each such process also corresponds to a *Keldysh operator* acting on *both* system and environment which is obtained by 'cutting' the group of Keldysh diagrams 'in halves' and summing them. These

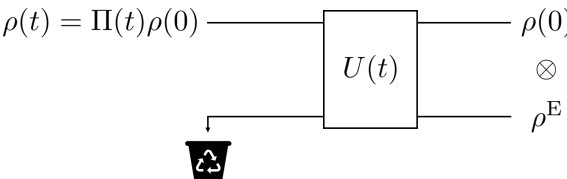

Figure 1: Open system evolution as a quantum circuit: The evolution $\Pi(t)$ is obtained by tracing out the environment after joint unitary evolution. The trashcan indicates the inaccessible or discarded information about the state of the environment.

Keldysh operators turn out to be the central objects of interest: They produce the Kraus operators – acting *only* on the system – by Wick normal-ordering and taking environment matrix elements.

Sec. 4 builds on this partial 'undoing' of the environment trace which is a convenient way to see the connection between microscopic Keldysh diagrams and Kraus operators. Focusing on the *generators* of infinitesimal time-evolution in either approach, we derive two equivalent hierarchies of time-nonlocal evolution equations. The hierarchy for the Keldysh operators presents a useful exact unraveling of the original Schrödinger equation for system plus environment. It *explicitly* encodes the operator-sum structure of the reduced dynamics *before* tracing out the environment. The result of this analysis is the explicit functional dependence of the memory kernel superoperator $\Sigma = \sum_m \Sigma_m[\sigma_0, \ldots, \sigma_m]$ in Eq. (3) on the self-energy operators $\sigma_0, \ldots, \sigma_m$ generating the Keldysh operators in Eq. (2). It thus expresses the *operator-sum theorem* for general *infinitesimal* reduced evolutions and reveals at the deepest level that CP and TP appear as incompatible constraints because of the difference between *time-ordering* of the microscopic *couplings* on one and two branches of the Keldysh real-time contour, respectively.

Then, in Sec. 5 we illustrate the generally applicable formalism for the simplest possible model of a single fermionic mode interacting with a hot continuum of fermion modes. This Markovian example only serves to highlight the inner workings of the approach and physical principles involved, delegating the technically much more demanding applications to future work.

Finally, in Sec. 6 we give both a summary of the technical results and further comment on the duality of CP vs. TP approximations tied to quantum information and statistical field theory, respectively.

## 2 Quantum-information approach to reduced dynamics

As emphasized above, quantum information clearly defines physical processes in an operational way, accounting in particular for the role of entanglement. To obtain such an operational formulation we express the evolution (1) in terms of a quantum circuit as depicted in Fig. 1. Taking the partial trace amounts to trashing all information about the environment, i.e., averaging over all possible measurement outcomes. The coupling $V(t)$ between system and environment generating the unitary evolution $U(t) = T \exp\left[-i \int_0^t d\tau V(\tau)\right]$ correlates the system with the environment over time. The above mentioned microscopic models in which the environment consists of reservoirs with a *continuum* of modes in *mixed thermal states* and where particles on the system strongly *interact*, however, preclude direct evaluation of the dynamical map (1). In the following we show how the key result (2) of quantum information can be combined with approaches based on Eq. (3) that aim calculate this nontrivial map.

## 2.1 Entanglement, dynamics and complete positivity

The dynamics (1) of interest has the important property that when applied *in the presence of entanglement*, it still produces physical states. One can show that the worst-case scenario is encountered in the evolution of the system when it is maximally entangled with a non-evolving copy of itself. In this case, applying $\Pi \otimes \mathcal{I}$ to the maximally entangled state $|\mathbb{1}\rangle = \sum_k |k\rangle \otimes |k\rangle$, see App. C, results in an operator

$$\mathrm{choi}(\Pi) = \left(\Pi \otimes \mathcal{I}\right)|\mathbb{1}\rangle\langle\mathbb{1}|. \tag{7}$$

This so-called Choi-Jamiołkowski operator is positive if and only if [46,71] the dynamics has the form (1) which in turn is equivalent [42] to the operator-sum form (2). The *complete positivity* (CP) of the evolution $\Pi$ ensures that the operator $\mathrm{choi}(\Pi)$ is a well-defined physical state (when trace-normalized) *and vice-versa*. Preserving the CP property of the dynamics of interest (1), i.e., its Kraus form (2), is therefore required to correctly account for *dynamics in the presence of entanglement*. It is well-known that many [57] maps $\rho(0) \mapsto \rho(t)$ may well preserve positivity (PP) of $\rho(0)$ but fail to do so in the presence of entanglement. Such PP maps that are not CP take a well-defined quantum state to an operator (7) with some strictly negative eigenvalues and cannot be implemented in a quantum circuit, i.e., they are unphysical in the sense of non-operational. Even if one is sure that an approximation gives a PP map, it may thus still not correctly account for entanglement[5] by failing to be CP.

## 2.2 Purification of the mixed environment state

Despite its clarity, the above standard formulation of the dynamics (1) is admittedly useless for many open-system approaches addressing nontrivial microscopic models. To arrive at a more compatible formulation, the central idea is to explicitly keep track of the dynamical correlations between the system and its environment by extending the environment *and* the evolution map. A computational approach which makes this explicit at each step will manifestly guarantee the complete positivity of the evolution as we work out in detail for the the Keldysh diagrammatic approach in Sec. 3.

We first need to distinguish correlations between the system S and the environment E built up over time from the correlations already present in the initial, mixed environment state written in its eigenbasis as $\rho^{\mathrm{E}} = \sum_n \rho_n^{\mathrm{E}}|n\rangle\langle n|$. The standard way to achieve this in quantum information theory is by purifying the environment state, i.e., expressing it as $\rho^{\mathrm{E}} = \mathrm{Tr}_{\mathrm{E}'}|0\rangle\langle 0|$, where

$$|0\rangle \equiv \sum_n \sqrt{\rho_n^{\mathrm{E}}}|n\rangle \otimes |n\rangle \tag{8}$$

denotes the *purified environment state*. This is a pure, entangled state in a doubled[6] environment space EE$'$ written in terms of the tensor product of the eigenbasis $\{|n\rangle\}$ of $\rho^{\mathrm{E}}$. We already note that this purification is important for the analysis but will not need to be computed in the end. We can bring Eq. (1) into the equivalent form

$$\Pi(t) = \mathrm{Tr}_{\mathrm{EE}'} W_{\mathrm{EE}'}(t) \bullet W_{\mathrm{EE}'}^\dagger(t), \qquad W_{\mathrm{EE}'}(t) := \left(U(t) \otimes \mathbb{1}_{\mathrm{E}'}\right)|0\rangle, \tag{9}$$

where we note that the ancilla E$'$ is not affected by unitary evolution. Due to unitarity of $U$, the new map $W_{\mathrm{EE}'} : \mathrm{S} \to \mathrm{S} \otimes \mathrm{E} \otimes \mathrm{E}'$ is an isometry, $W_{\mathrm{EE}'}^\dagger W_{\mathrm{EE}'} = \mathbb{1}_{\mathrm{S}}$, which together with the

---

[5] Such PP maps *mathematically* play a central role precisely because their 'failure' detects the entanglement of mixed quantum states at their input. The corresponding hermitian (non-positive) Choi-Jamiołkowski operators are *physical observables* (not states). Measuring a negative expectation value of such an 'entanglement witness' is nowadays commonly used to experimentally detect entanglement.

[6] Doubling the system is sufficient to allow any mixed state to be purified.

$\rho(t) = \Pi(t)\rho(0)$ — $U(t)$ — $\rho(0)$     $W_{\mathrm{EE}'}(t)$     $K^e(t)\rho(0)K^{e\dagger}(t)$ — $U(t)$ — $\rho(0)$

Figure 2: Quantum circuits corresponding to the operator sum (10) and its individual terms. (a) Reduced evolution: Evolved state is averaged over all *possible* outcomes $e$ of measurements on the purified environment indicated by the trash can. (b) Conditional evolution: Classical communication of a specific outcome $e$ of a measurement results in a state update described by a single Kraus operator.

form (9) is necessary and sufficient for $\Pi$ to be a CP-TP superoperator [42, 72]. This ensures it corresponds to some quantum circuit built using only unitary components, pure states and projective measurements. In Fig. 2(a) we show this quantum circuit for the form (9) in which $W_{\mathrm{EE}'}$ is indicated by the shaded part.

Any change of the purified environment state $|0\rangle$ is now certain to be caused by the microscopic coupling $V$. This becomes explicit when we expand the trace in Eq. (9) in terms of some orthonormal basis $\{|e\rangle\}$ for $\mathrm{EE}'$ giving the operator-sum representation (2):

$$\Pi(t) = \sum_e K^e(t) \bullet K^{e\dagger}(t), \qquad K^e(t) = \langle e|W_{\mathrm{EE}'}(t) = \langle e|U(t) \otimes \mathbb{1}_{\mathrm{E}'}|0\rangle. \tag{10}$$

When a measurement on this extended environment does not find it in its known original state, $|e\rangle \neq |0\rangle$, this directly probes the entanglement generated between the system and the original environment E. Accordingly, the evolution conditioned on this outcome is $K^e(t)\rho(0)K^{e\dagger}(t)$ as shown in Fig. 2(b). We stress that for the *original* environment E, due to its mixed nature (e.g. thermal noise), it is not possible to keep track of changes incurred by the evolution only from measurements at the final time $t$.

This train of thoughts is of course well known from quantum information theory. However, tangible means to obtain the Kraus operators (10) from the full unitary $U \otimes \mathbb{1}_{\mathrm{E}'}$ acting on a *continuous* environment in practice require field-theoretical methods. There, it is easier to evaluate *Wick-averages* of the unitary $U$ after it has been expanded in the microscopic coupling $V$, and, furthermore, to consider *infinitesimal* evolutions using kinetic equations. In the following, we will explicitly provide the connection to this field-theoretical approach allowing one to preserve complete positivity in a framework better suited for approximate treatments.

## 2.3 Purification of the evolution operator

To do so, the picture of the measurement-conditioned evolution needs to be further refined: The above introduced states $|e\rangle$ on the purified environment do not account for the *internal structure of the evolution $U$ relative to the environment state* $\rho^{\mathrm{E}}$. This is revealed when expanding the unitary into normal-ordered components as discussed in App. B:

$$U(t) = k_0 + \sum_{m \neq 0} :k_m(t): \quad . \tag{11}$$

The normal-ordering indicated by $:\,:$ is a standard procedure reviewed in App. A. It is generally applicable, i.e., it does not only apply to noninteracting environments that we will consider later on. The normal-ordered components of the evolution are labeled by $q = 1, 2, \ldots$ environment modes $(m_q \ldots m_1) := m$ which have interacted with the system by a term $:k_m:$, see Eq. (6) ff. for examples and Sec. 5. We indicate by $m \neq 0$ that we sum over all of these. By

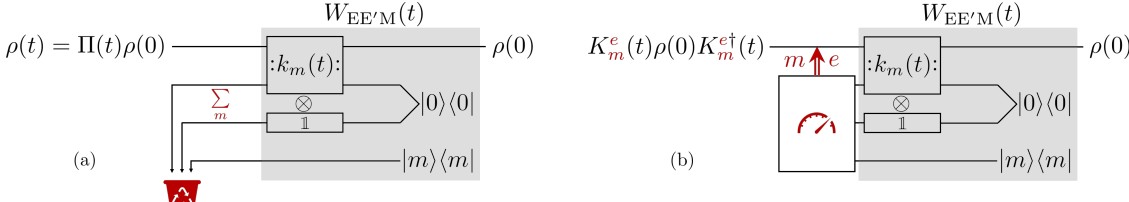

Figure 3: Quantum circuits corresponding to the refined operator-sum (14) and its terms. The auxiliary meter M keeps track of *which* modes $m$ of the environment have interacted with the system. (a) Reduced evolution: Neither the outcomes of *possible* measurements on the purified environment, $e$, nor those on the auxiliary meter, $m$, are communicated. (b) Conditioned evolution: The list of modes $m$ that have interacted with the system *and* the measured states $|e\rangle$ are communicated.

construction, the operator $k_0$ is exceptional in that it acts trivially on the environment and the label $m = 0$ indicates 'no modes'. It is important to note that the normal-ordering $::$ need not be evaluated in the end but continues to play a simplifying role: It ensures by construction that tracing out the environment in

$$\operatorname*{Tr}_{E} :k_{m'}^{\dagger}(t): :k_m(t): \rho^{E} \propto \delta_{m'm} \tag{12}$$

gives a nonzero system operator only if the modes $m$ and $m'$ match.

We can exploit the structure (11) from the start by explicitly keeping track of the labels of the environment modes $m$ using a superoperator $W_{EE'M} : S \to S \otimes E \otimes E' \otimes M$ which stores these in a register $|m\rangle$ in an auxiliary *meter* space M. This leaves the total evolution unaltered if we additionally trace out the meter:

$$\Pi(t) = \operatorname*{Tr}_{EE'M} W_{EE'M}(t) \bullet W_{EE'M}^{\dagger}(t), \qquad W_{EE'M}(t) := \sum_m \big( :k_m(t): \otimes \mathbb{1}_{E'} \big)|0\rangle \otimes |m\rangle. \tag{13}$$

Due to the normal-ordering property (12) the new map is also an isometry, $W_{EE'M}^{\dagger} W_{EE'M} = \mathbb{1}_S$, and can thus be regarded as another purification[7] of the original evolution map (1). Expanding both the trace over M and EE' we obtain a refined operator-sum

$$\Pi(t) = \sum_{m,e} K_m^e(t) \bullet K_m^{e\dagger}(t), \qquad K_m^e(t) := \langle e| \otimes \langle m| W_{EE'M}(t) = \langle e| :k_m(t): \otimes \mathbb{1}_{E'}|0\rangle, \tag{14}$$

where by measuring the meter's register we can pick out the normal-ordered component $:k_m:$ of the evolution. Here, the *Keldysh*[8] operators $k_m$ are still operators on system and environment, in contrast to the *Kraus operators* $K_m^e(t)$ which act on the system only. Keeping the environment E via the $k_m$ is at first a technical trick to avoid writing out matrix elements. Later on it will allow for an interesting unraveling of the Schrödinger equation of system plus environment [Sec. 4.2]. By construction, for $m = 0$ the Keldysh operator factorizes as $k_0 = K_0^0 \otimes \mathbb{1}_E$ and the only remaining Kraus operator

$$K_0^e = \langle e|0\rangle \operatorname*{Tr}_{E} U \rho^{E} = \delta_{e,0} \langle U \rangle_E \tag{15}$$

is the environment average of the unitary $U$.

---

[7]The purification of the isometry $W_{EE'} \bullet (W_{EE'})^{\dagger} = \operatorname{Tr}_M W_{EE'M} \bullet (W_{EE'M})^{\dagger}$ corresponds to altering Fig. 2 to Fig. 3 without changing the input-output relation of the circuit. Compare this formula with the mixed-state purification $\rho^E = \sqrt{\rho^E}\sqrt{\rho^E} = \operatorname{Tr}_{E'} |0\rangle\langle 0|$ that we used earlier. Eq. (9) and (13) are both purifications (Stinespring dilations) of the same CP-TP superoperator (1).

[8]This nomenclature will be motivated in Sec. 3.

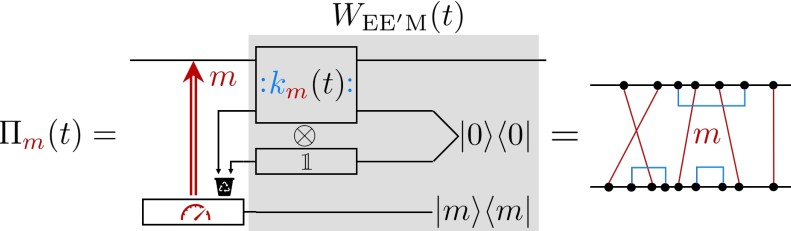

Figure 4: *Partially* conditioned evolution (16b): the list of modes *m that* have interacted with the system is communicated (but not the measurement outcomes). The key result Eq. (21b), explained in Fig. 6, demonstrates that this circuit corresponds to the indicated Keldysh diagrams with fixed modes on the red contractions summed over all possible blue contractions.

The circuit in Fig. 3 summarizes the above: Purifying the environment to an entangled state $|0\rangle$ (to eliminate thermal noise) and purifying the unitary $U$ to an isometry $W_{\mathrm{MEE}'}$ (using normal-ordering) results in the refined operator-sum (14) for $\Pi$. The trace over the purified environment and the meter correspond to discarding the outcomes $m$ (mode labels) and $e$ (state of purified environment) by summing over them.

Importantly, we can now choose the level of refinement: if one only resolves the trace over the meter space M one may write the original time-evolution problem as

$$\Pi(t) = \mathrm{Tr}_{\mathrm{E}} \, U(t) \Big[ \bullet \otimes \rho^{\mathrm{E}} \Big] U^{\dagger}(t) = \sum_m \mathrm{Tr}_{\mathrm{E}} \, {:}k_m(t){:} \Big[ \bullet \otimes \rho^{\mathrm{E}} \Big] {:}k_m^{\dagger}(t){:} \quad . \tag{16a}$$

This is the key relation of the paper: in Sec. 3 we show that this exactly corresponds to calculations employing Keldysh real-time diagrams and Wick's theorem. There the *Keldysh* operators $k_m$ (acting on the system and environment) are uniquely represented by groups of Keldysh diagrams for a *conditional* propagator (acting on the system only),

$$\Pi_m = \mathrm{Tr}_{\mathrm{E}} \, {:}k_m(t){:} \Big[ \bullet \otimes \rho^{\mathrm{E}} \Big] {:}k_m^{\dagger}(t){:} \quad , \tag{16b}$$

depending on the measurement outcome $m$. This corresponds precisely to the modified quantum circuit as shown in Fig. 4 and explicitly ensures that $\Pi_m$ is a CP superoperator. It is these objects that one should calculate while formally keeping track of $m$ in order to obtain a CP result for $\Pi = \sum_m \Pi_m$.

We remark that the above purification approach can be completely avoided by directly working in Hilbert-Schmidt or Liouville-space [64] which is useful for, e.g., deriving projection-based approaches that guarantee CP evolution [44]. The above quantum-information approach, however, emphasizes the clear *operational* test: If a complicated mathematical expression can be expressed as a physically implementable quantum circuit without initial system-environment correlations, then it *must* be a CP superoperator.

## 2.4 Trace preservation (I): State-evolution correspondence

So far we have focused on the CP property which is almost trivialized in the quantum-information approach: when $\Pi$ is written in the operator-sum form it can be verified term-by-term. However, for the TP property of the evolution this implies the opposite: It requires a nontrivial sum of quadratic operator expressions

$$\sum_{m,e} K_m^{e\dagger}(t) K_m^e(t) = \mathbb{1}_{\mathrm{S}}, \tag{17a}$$

to match the system identity operator at all times $t$. For the Keldysh operators this condition translates to

$$\sum_m \underset{\text{E}}{\text{Tr}} :k_m^\dagger(t)::k_m(t):\left(\mathbb{1}_\text{S} \otimes \rho^\text{E}\right) = \sum_m \Pi_m^\dagger(t)\mathbb{1}_\text{S} = \mathbb{1}_\text{S}. \qquad (17\text{b})$$

This implies that the conditional propagators $\Pi_m$ are trace-nonincreasing (TNI) superoperators, $\text{Tr}\,\Pi_m \le \text{Tr}$. Keeping in mind applications with continuous environments these are highly nontrivial[9] conditions to check or guarantee.

In this respect, the relation of $k_m$ to the Keldysh real-time expansion announced at Eq. (16) is intriguing: in the latter approach TP is a trivial condition to satisfy whereas CP is nontrivial to guarantee. There is thus a fundamental incompatibility in the practical requirements for ensuring CP and TP, no matter in which form one writes the propagators $\Pi$ and $\Pi_m$. This CP-TP duality was noted in Ref. [70] and is in fact an expression of the fundamental correspondence between states and evolutions [73] based on the isomorphism of de Pillis [74], Jamiolkowski [75] and Choi [76]. This general point of view is expanded in App. C but we will instead focus on the way this CP-TP duality is expressed on the microscopic level of diagrammatic contributions that one encounters in practice when we return to this issue in Sec. 3.4.

# 3 Statistical field-theory approach to reduced dynamics

The considerations of the previous section hold generally –noting the key assumption of an initially uncorrelated system and environment. In the following, we will focus on the Keldysh real-time diagrammatic technique [19] which is a powerful tool for calculating density-operator evolutions beyond the weak-coupling limit [21–23, 26] when the environment is a composite of noninteracting, continuous reservoirs of fermions or bosons, each in a different thermal state. We note, however, that most of our conclusion can be generalized [64]. Our aim is to reorganize this well-established diagrammatic expansion such that one can easily identify the Keldysh- and Kraus operators in terms of practical Wick-averages. This makes our approach of immediate relevance for various formulations of approximations [26–28, 32] in terms of such diagrams.

## 3.1 Standard Keldysh real-time expansion

The Keldysh real-time diagrammatic approach is based on the formal *expansion in the coupling* $V$ of the unitary $U(t) = T \exp\left[-i \int_0^t d\tau V(\tau)\right]$ on both sides of Eq. (1) which is necessary to be able to perform the Wick averages and explicitly integrate out the reservoirs. All contributions are represented by diagrams of the type shown in Fig. 5. These keep track of whether a coupling term $V$ stems from the evolution $U(t)$ [left of $\rho^\text{E}$ in (a), upper branch of the Keldysh time-contour in (b)] or from the adjoint evolution $U^\dagger(t)$ [right of $\rho^\text{E}$, lower branch of the contour]. As indicated in the caption, the two ways of drawing the same diagram have distinct advantages needed later on.

Wick's theorem expresses the environment trace as the sum of products of pair-contractions of environment field operators appearing in the *different* $V$'s. These are drawn as connecting lines in Fig. 5 for the example of a bilinear coupling. We assume that the operator $V$ has been normal-ordered with respect to the initial environment state, cf. App. A. Thus, any partial

---

[9]Of course, *both* CP and TP are trivial from the original formulation of the problem (deriving from the unitary of $U$). This is, however, irrelevant if one cannot compute $U$. In practice, approximations that might violate CP or TP are made in terms of the various quantities that we discuss throughout the paper, be it $\Pi$, $\Sigma$, $\Pi_m$, $\Sigma_m$, $k_m$, $\sigma_m$ or $K_m^e$.

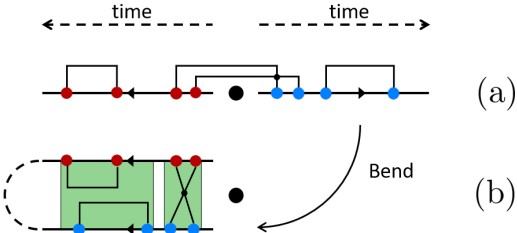

Figure 5: Two equivalent ways of drawing Keldysh real-time diagrams. (a) Nonstandard left-right form: Time runs outward from the center. This is useful for identifying the operator-sum form of the real-time expansion, cf. Fig. 6. (b) Standard forward-backward form, obtained by bending the right branch in (a) backward: Time runs from right to left to agree with the order of writing expressions. This form is advantageous for identifying the diagrammatically irreducible components (green) of the time-evolution in Sec. 4. For fermions, the sign of the contribution of a diagram is given by $(-1)^{n_c}$ where $n_c$ is the number of crossings of the environment contraction lines, which is the same for (a) and (b). We note that it is common to denote the adjoint $U^\dagger(t) = U(-t)$ the 'backward' evolution, imagining to traverse the contour first forward along the upper branch, then across the dashed connector 'backward' on the lower branch. Here, this is misleading: Time always runs in the forward direction, also on the lower contour as indicated in (b).

contraction of fields appearing in the same vertex $V$ gives zero[10]. Any nonzero environment average of the coupling is thereby already absorbed into the definition of the system Hamiltonian $H(t) + \langle V \rangle_E(t) \to H(t)$. Rules for translating diagrams to expressions are summarized in App. D but the following considerations do not require these details, the diagrammatic representations suffice. More model-specific diagrammatic rules [21–23, 26, 77] and their equivalent formulations [24, 34, 37, 41] make this a very efficient technique for higher-order calculations [19, 21–23, 25, 27, 28, 77].

From the resulting diagrammatic series one infers the self-consistent Dyson equation,

$$\Pi = \pi + \pi * \Sigma * \Pi, \tag{18}$$

using standard field-theoretical considerations: the self-energy kernel $\Sigma$ is defined by the sum of two-branch-irreducible diagrams [26]. For this distinction one needs to draw Keldysh diagrams in the standard form shown in Fig. 5(b): such a diagram is (ir)reducible when it can(not) be split up by a *vertical cut* without hitting a contraction line. Taking the time-derivative of the Dyson equation (18) gives the time-nonlocal kinetic equation (3) noting that $\pi = \mathcal{I}$ in the interaction picture.

A problem of the Keldysh diagrammatic expansion in the above standard form is that it does not explicitly exhibit the structure of an operator-sum (14) which it must have by the CP property of the original problem (1). This unknown structure complicates the formulation of approximations that are often made by omitting and resumming diagrams: By doing so one may neglect contributions that are essential for the operator-sum structure to guarantee CP evolution. Moreover, such approximations are frequently made on the level of the self-energy $\Sigma$ which inherits an even more complicated structure [Sec. 4.3] from the operator-sum form of the propagator $\Pi$ considered so far.

---

[10]Normal-ordering of the coupling $V$ does not imply that the *unitary* $U$ that it generates is normal-ordered: A sequence of normal-ordered $V$'s contributing to $U$ still allows for nonzero partial contractions.

## 3.2 Reorganized Keldysh real-time expansion: Cutting and pasting rules

We therefore reorganize the diagrammatic series guided by the quantum-information considerations of Sec. 2 and identify which *groups* of diagrams rigorously correspond to a *physical* measurement processes. We write the normal-ordering expansion (11) as

$$U(t) = k_0 + \sum_{q=1}^{\infty} \sum_{m_q \dots m_1} :k_{m_q \dots m_1}(t): \quad , \tag{19}$$

which is constructed in App. B in detail. Here, $k_{m_q \dots m_1}$ is the evolution $U$ partially contracted such that only the modes labeled by $m = m_q \dots m_1$ remain. We ignore the ordering of the modes, i.e., $k_{m_q \dots m_1}$ is invariant under mode permutations and the summation $m_q \dots m_1$ is restricted so as not to double count terms. The special case $k_0$ denotes the part of $U$ in which no environment operators are left, i.e., the *environment average*

$$k_0 := \langle U \rangle_{\mathrm{E}}. \tag{20}$$

Our central formula (16) explicitly ensures that the CP-TP evolution of interest,

$$\Pi(t) = \sum_{q=0} \sum_{m_q \dots m_1} \Pi_{m_q \dots m_1}(t), \tag{21a}$$

decomposes into superoperators

$$\Pi_{m_q \dots m_1} = \underset{\mathrm{E}}{\mathrm{Tr}} :k_{m_q \dots m_1}(t): \left[ \bullet \otimes \rho^{\mathrm{E}} \right] :k^{\dagger}_{m_q \dots m_1}(t):, \tag{21b}$$

which are CP-TNI *because* they are *conditional* evolutions depending on measurement outcomes $m_q \dots m_1$ indicating which environment modes the system has interacted with. In Fig. 6 we explain how each $\Pi_{m_1 \dots m_q}$ corresponds to a precise *group* of Keldysh diagrams. In (a)-(b) we first sketch how the standard Keldysh expansion is obtained using the now more pertinent left-right form of diagrams introduced in Fig. 5(a). In Fig. 6(c) we show how the diagrammatic elimination of the environment E (stage 1) and the meter M (stage 2) corresponds to Wick contracting [Eq. (21b)] and summing over modes $m_q \dots m_1$ [Eq. (21a)], respectively.

Thus, the *elementary physical processes* contained in the standard real-time expansion for the propagator $\Pi$ when computed in terms of Wick contractions are the conditional propagators $\Pi_{m_q \dots m_1}$. However, once the mode-sums for *these* contractions have been performed the CP property is hidden and approximations may inadvertently break up the CP structure. We now discuss the two computational stages that the above implies.

**Stage 1, Eq. (21b)** First, the conditional propagator $\Pi_m$ needs to be computed by eliminating E. This can be done in two equivalent ways:

(A) Compute $\Pi_{m_q \dots m_1}$ directly. An advantage of our approach as compared to Refs. [43,44, 62,63] is that it does not necessarily require a fundamentally new formulation: The standard rules [see App. D] of the existing Keldysh approach merely have to be complemented with a diagrammatic CP-rule which instructs one to keep track of $m_q \dots m_1$:

*Rule for the conditional propagator $\Pi_{m_q \dots m_1}$: Sum all diagrams that have a fixed number q of contractions between the different branches of the Keldysh contour with a fixed set of mode indices $m_q \dots m_1$ assigned to these lines in every possible order. This includes summing all possible intra-branch contractions separately and summing over the mode indices of these contractions. Perform all two-branch ordered time integrations.*

Note that this rule requires *both* reducible and irreducible diagrams in the two-branch sense to be summed. In Fig. 7(a) we illustrate why this is technically important and make explicit the intricate difficulty of relating time-ordering on one and two branches of the Keldysh

$$\Pi(t) = \sum_{n,n'} \mathrm{Tr}_{\mathrm{E}} \quad \text{———•—•—•—•—} \quad \rho^{\mathrm{E}} \bullet \quad \text{•—•—•—•—•—•—•—•—•→} \quad \text{(a)}$$

$$= \sum_{n,n'} \sum_{\mathrm{diagr.}} \sum_{\{m_i\}} \quad \bullet \quad \text{(b)}$$

$$= \sum_{\mathrm{diagr.}} \left( \sum_{n} \right) \bullet \left( \sum_{n'} \right)$$
$$+ \sum_{\mathrm{diagr.}} \sum_{q=1}^{\infty} \sum_{\{m_i\}} \left( \sum_{n} \right) \bullet \left( \sum_{n'} \right) \quad \text{(c)}$$

Figure 6: Reorganization of the standard real-time expansion in terms of left-right Keldysh diagrams [cf. Fig. 5]: (a) Expansion of the unitary $U$ [$U^\dagger$] into vertices $-iV$ in red [$+iV$ in blue]. (b) Evaluation of the trace gives both inter- and intra-branch contractions (c) Two-stage reorganization of (b). Stage 1: Sum up *all* intra-branch contractions separately (round brackets), but restrict the number of inter-branch contractions to $q = 0, 1, 2, \ldots$ fields and fix their modes $m_q \ldots m_1$. Stage 2: Sum up all contributions from modes $m_q \ldots m_1$ for all $q$.

contour. To obtain $\Pi_m(t)$ we sum up terms on the right which are single functions of the final time $t$ obtained by two-branch time-ordered integrations. If the CP-rule is obeyed, this sum is able to produce a single term that factorizes[11] into two separate functions of time, $\mathrm{Tr}_{\mathrm{E}}\,{:}k_m(t){:}({\bullet} \otimes \rho^{\mathrm{E}}){:}k_m^\dagger(t){:}$. Here, the time-integrations are performed independently *inside* $k_m$ on each branch separately. Note that this 'completing of the square' of ordered time-integrations is not possible when summing two-branch irreducible diagrams only, and the latter can therefore never correspond to physical processes [Eq. (29)].

(B) Compute $\Pi_{m_q \ldots m_1}$ from $k_{m_q \ldots m_1}$. Alternatively, one can evaluate the right hand side of Eq. (21b). The required Keldysh operators can be obtained from the value of a Keldysh diagram that is cut in half, i.e., from a single time-branch.

*Cutting rule for $k_{m_q \ldots m_1}$: Sum all half-diagrams that have q of external fields acting on a fixed set of environment modes $m_q \ldots m_1$ over which we symmetrize. This includes summing over all possible intra-branch contractions and their mode indices. Perform all one-branch time-ordered integrations, including the times of the external vertices.*

The full diagrammatic rules for calculating the Keldysh are summarized in App. E and are conveniently similar to the standard Keldysh rules [App. D]. The required value for $\Pi_{m_q \ldots m_1}$ is then obtained by pasting two halves together as follows:

*Pasting rule for $\Pi_{m_q \ldots m_1}$: Fully contract the external vertices appearing on both sides of* $\mathrm{Tr}_{\mathrm{E}}\,{:}k_{m_q \ldots m_1}(t){:}({\bullet} \otimes \rho^{\mathrm{E}}){:}k_{m_q \ldots m_1}^\dagger(t){:}$ *in all possible ways.*

We note that the normal-ordering in Eq. (21b) is only an instruction to simplify the evaluation by discarding all internal Wick contractions of $k_m$ and $k_m^\dagger$. This is in contrast to the Kraus operators[12] $K_m^e = \langle e|({:}k_m{:} \otimes \mathbb{1})|0\rangle$ for which the normal-ordering has to be explicitly evaluated. It should also be noted that when using Wick contractions, the most elementary physical process $\Pi_m$ is *never* described by a single or even a finite number of Keldysh real-time diagrams. Typically, guaranteeing CP in approximate dynamics calls for methods that are nonperturbative in the system-environment coupling.

---

[11]Such factorization brings computational speed up in higher-order perturbative calculations [23] and is also necessary to simplify the noninteracting limit, see also page 19 of Ref. [24].

[12]In Refs. [43,62,63] Kraus operators are considered which depend on the internal times of the coupling vertices. In our Kraus operators $K_m^e$ we instead perform all internal time-integrations and directly work with the underlying Keldysh operators $k_m$ which are not yet restricted by normal-ordering.

$$\int \frac{\overset{\tau_+ \ \ \tau'_+}{\rule{1cm}{0pt}}}{\underset{\tau_- \ \ \tau'_-}{\times}} \ = \ \int \frac{\overset{\tau_+ \ \ \tau'_+}{\rule{1cm}{0pt}}}{\underset{\tau_- \ \ \tau'_-}{\rule{1cm}{0pt}}} + \int \frac{\overset{\tau_+ \ \ \tau'_+}{\rule{1cm}{0pt}}}{\underset{\tau_- \ \ \tau'_-}{\rule{1cm}{0pt}}} + \int \frac{\overset{\tau_+ \ \ \tau'_+}{\rule{1cm}{0pt}}}{\underset{\tau_- \ \ \tau'_-}{\rule{1cm}{0pt}}} + [\ \mathrm{Swap} \updownarrow\ ] \quad (a)$$

Figure 7: Cutting and pasting rules for Keldysh diagrams to test and guarantee CP, respectively. (a) *Time ordering*: Factorization into separate one-branch time-ordered expressions (left) –dictated by the operator-sum form– requires both two-branch reducible and irreducible Keldysh diagrams to be summed (right). (b) *Cutting*: Given a selection of diagrams on the *right* hand side, one can verify it has an operator-sum form by first cutting *horizontally* and summing up the halves. *Pasting*: Subsequently pasting the halves back together should recover the selection. Leaving out, for example, the center two diagrams on the right hand side, this test of 'completing the square' will fail. When directly selecting *half*-diagrams on the *left* hand side, every possible approximation (select only green, only blue or select both) by construction produces one of the three groups indicated by dashed lines and shading on the right, each of which is a CP superoperator.

**Stage 2, Eq. (21a)**   It remains to sum the computed $\Pi_m$ over the modes[13] $m_q \dots m_1$ and their number $q = 0, 1, \dots$ in order to eliminate the meter M. Our example in Sec. 5 will illustrate that truncating $q$, the number of modes, in stage 2 correlates with the time to reach the stationary state relative to the time-scale that is set by the time-dependence of $\Pi_{m_q \dots m_1}(t)$ obtained in stage 1 [Eq. (50)]. Because the expansion index $m_q \dots m_1$ is physically motivated, the truncation of stage 2 never jeopardizes CP.

### 3.3   CP approximations

The above transparent rules translate the operator-sum theorem to the framework of the standard real-time expansions on one and two branches of the Keldysh time-contour. Above all, this provides explicit means of translating problems and solutions between the very different formulations employed in quantum information theory and statistical physics on the microscopic level of the system-environment coupling $V$.

As expected, evaluating the objects obtained by these simple rules is by no means easy. The importance of our general framework is that at least the large variety of existing approximations can be formulated, compared and systematically improved. In Sec. 4 we make a further step towards practical applications by deriving useful self-consistent and infinitesimal evolution equations. The reorganized expansion [Eq. (21), Fig. 6] already shows how *CP approximations* can be constructed generically. Fig. 7(b) illustrates this for approximations based on selecting *complete* Keldysh diagrams[14]. In order to guarantee CP, such a selection must pass the test of *horizontally cutting* the Keldysh diagrams using their standard representation [Fig. 5(b)] and summing up their halves to obtain $k_m$ [left hand side in Fig. 7(b)]. By the pasting rule, one can check that $k_m$ reproduces the groups of two-branch diagrams originally selected [right

---

[13]This list may include repetitions, e.g., for an environment with three modes, $m = (112333)$ is included.

[14]We note that this is not the only way to construct CP approximations: The pasting rule can be modified to ensure both CP and TP. This is effectively what is done in Refs. [62, 63], see also Sec. 3.5.

$$\operatorname{Tr}\Pi(t) = \operatorname{Tr} \;\rule{1cm}{0pt}\; + \sum_{\text{Rest}} \;\underset{iV}{\overset{-iV}{\cdots}}\; + \;\cdots\; \qquad \text{(a)}$$

$$= \operatorname{Tr}\left[\;\rule{0.8cm}{0pt}\; + \ldots + \;\cdots\; \right] + \left[\;\cdots\; + \ldots + \;\cdots\; \right] + \left[\;\cdots\; + \ldots\right] \qquad \text{(b)}$$

Figure 8: (a) Standard real-time expansion: trace preservation is guaranteed by the vertex flipping rule (22). (b) Reorganized expansion: Brackets indicate groups of diagrams that belong to the same physical process generated by one Keldysh operator $k_m$. Applying the simple rule of (a) reveals a nontrivial network of cancellations between diagrams in different brackets corresponding to different Keldysh operators. This shows that CP and TP cannot be ensured simultaneously by selection of complete $\Pi_m$-groups of diagrams.

hand side in Fig. 7(b)]. This amounts to checking that one can diagrammatically 'complete the square' to the operator sum (21b).

Clearly, directly selecting groups of *two-branch* diagrams, each group corresponding to one conditional propagator $\Pi_m$, passes this test. This amounts to selecting *physical* processes, i.e., possible measurements performed on the meter M, and *therefore* preserves CP. This is illustrated by the example later on [Eq. (52)]. The approximate propagator is a TNI[15] super-operator [Eq. (17)] expressing that some probability was lost by omitting physical processes.

A more advanced scheme is to directly select complete *one-branch* diagrams for the Keldysh operators $k_m$ [left-hand side in Fig. 7(b)]. One can then infer which distinct two-branch diagrams they generate and equivalently select only these in the standard Keldysh expansion. This amounts to a modification of the propagator $\Pi_m$ by selecting a subset of diagrams from that group. Because this is done without breaking the CP operator-sum structure, the modified contribution still corresponds to a physical process occurring with a positive probability, cf. Sec. 3.5, *provided* the superoperators are TNI which is not automatic[15].

### 3.4 Trace preservation (II): Microscopic state-evolution correspondence

We now address the constraints imposed by TP [Sec. 2.4] and discuss the apparent duality with the constraints imposed by CP on the microscopic level of the coupling $V$.

In Fig. 8(a) we first show how in the standard Keldysh real-time diagrammatic approach TP is ensured by pairs of diagrams with the latest (leftmost) coupling vertex $V$ appearing on opposite branches of the Keldysh contour. When taking the system trace, the cyclicity of the total trace allows us to move these latest vertices to either side,

$$\int dt_n \int dt'_{n'} \operatorname{Tr}[-iV(t_n)] \bullet iV(t'_{n'}) = -\int_{t'_{n'} < t_n} dt_n dt'_{n'} \operatorname{Tr} \bullet [iV(t'_{n'})][iV(t_n)]$$

$$-\int_{t'_{n'} > t_n} dt_n dt'_{n'} \operatorname{Tr}[-iV(t'_{n'})][-iV(t_n)]\bullet, \qquad (22)$$

where $\bullet$ denotes the initial state as well as all earlier vertices from the $n$-th and $n'$-th order terms of $U$ and $U^\dagger$, respectively. The resulting terms are already contained in different orders $(n \pm 1, n' \mp 1)$ of the double expansion in powers of $V$ up to an opposite sign. Thus, all terms

---

[15]For superoperators $S_A\rho := \operatorname{Tr}_E A(\rho \otimes \rho^E)A^\dagger$ we have $\operatorname{Tr} S_A\rho \le \operatorname{Tr}(S_A + S_B)\rho$ but $\operatorname{Tr} S_A\rho$ can exceed $\operatorname{Tr} S_{A+B}\rho$ for $B = -A$.

cancel pairwise except the zeroth order $n = n' = 0$ term [first diagram in Fig. 8(a)] which guarantees that $\operatorname{Tr} \Pi = \operatorname{Tr}$. Notably, since the total order of $V$ is $n + n'$ for the canceling terms, TP can be guaranteed easily order-by-order in the coupling-expansion. On the other hand, CP can neither be inferred by inspecting just a few terms in the standard Keldysh expansion nor be guaranteed order-by-order in $V$.

In the reorganized Keldysh expansion the difficulty of guaranteeing CP and TP is reversed. Yet, we can make use of the diagrammatic rule (22) to reveal the nontrivial interrelation of the Keldysh (and Kraus) operators that is implied by the nonlinear constraint (17b) involving *all* Keldysh operators:

$$\mathbb{1}_S = \sum_{q=0}^{\infty} \sum_{m_q \ldots m_1} \operatorname*{Tr}_E \; {:}k^{\dagger}_{m_q \ldots m_1}{:}{:}k_{m_q \ldots m_1}{:} \bigl( \mathbb{1}_S \otimes \rho^E \bigr). \tag{23}$$

In Fig. 8(b) we sketch a group of two-branch diagrams for some process $\Pi_{m_q \ldots m_1}$ (indicated by the middle bracket). Consider the two possible situations for the latest (leftmost) vertex indicated in red. This vertex is either already contracted within the same branch (green); then, the TP rule (22) demands that we also take into account a corresponding diagram whose latest vertex lies on the opposite branch and thus contributes to a process $\Pi_{m_{q+1} \ldots m_1}$ with one additional external vertex (right bracket). Otherwise, the vertex is contracted with the opposite branch (blue); then, the TP rule (22) requires a term in the process $\Pi_{m_{q-1} \ldots m_1}$ with one less external vertex (left bracket). This implies that the TP condition for the Keldysh operators making up each process is guaranteed by the pairwise contribution of coupling vertices to 'adjacent' operators, schematically

$$k_{m_{q-1} \ldots m_1} \longleftrightarrow k_{m_q \ldots m_1} \longleftrightarrow k_{m_{q+1} \ldots m_1}. \tag{24}$$

Thus, the TP condition on the Keldysh and Kraus operators that appears nontrivial [Eq. (17)] is *microscopically* clear. These pair cancellations do not form a simple chain but rather a network correlating[16] all terms in the operator-sum $\Pi = \sum_m \operatorname{Tr}_E {:}k_m{:}(\bullet \otimes \rho^E){:}k^{\dagger}_m{:}$.

Importantly, the CP structure requires one to include all possible two-branch time-orderings of the individual vertices [cf. Fig. 7(a)]. In particular, we must include both terms in the middle bracket in Fig. 8(b) which then rigidly connects two independent TP constraints (22). Because this applies to every pair of diagrams in the middle bracket, all groups of diagrams connected to it by a TP constraint must be included as well. Continuing this argument to these other groups one finds that eventually *all* diagrams have to be included. This is the ultimate microscopic ground for the CP-TP duality as it appears in practical approximations, complementing the general considerations of App. C.

## 3.5 CP vs. TP approximations

Having understood the conflicting demands of CP and TP microscopically, we now again consider the two fundamental ways of approximating and discuss the implications of losing either CP or TP by again polarizing the discussion on purpose.

Consider first *CP approximations*, in particular, the scheme discussed in Sec. 3.3 where the approximated propagator $\Pi_{q_{\max}} = \sum_{q=0}^{q_{\max}} \sum_{m_q \ldots m_1} \Pi_{m_q \ldots m_1}$ is obtained by discarding terms with more than $q_{\max}$ inter-branch contractions. Physically / operationally, this means we account for at most $q_{\max}$ modes that have interacted with the system up to the considered time $t$. We note that any restriction of the summation of any complete CP-TP operator-sum produces a CP superoperator which is no longer TP. However, because this approximation discards *entire*

---

[16]In terms of the state-evolution correspondence this network realizes the correlations within the Choi-state $\operatorname{choi}(\Pi)$ which ensure that its partial state is maximally mixed, see the requirement (67) in App. C.

*physical processes* (not: partial contributions to these), the resulting superoperator is still trace-nonincreasing (TNI) [Eq. (17b)]. It therefore retains a well-defined operational meaning[17]. The TP error that this approximation incurs can be quantified by the trace distance of the operator-constraint (17b),

$$\left| \mathbb{1}_S - \Pi_{q_{max}}^\dagger (\mathbb{1}_S) \right|, \tag{25}$$

computed from the adjoint of the approximated evolution $\Pi_{q_{max}}^\dagger$ that one has in hand. This error will be monotonically reduced by increasing $q_{max}$ since the added terms are CP-TNI superoperators. Physically / operationally, this means we account for more modes that have interacted with the system up to the considered time $t$. One expects that more modes are required when the system interacts longer with the environment.

We furthermore note that a truncation of the sum $\Pi(t) = \sum_m \Pi_m(t)$ is *not* a naive short-time expansion[18] of $\Pi(t)$ because each contribution $\Pi_m(t)$ contains an infinite sum of intra-branch contractions [Fig. 6] causing a dissipative decay. This nontrivial time-dependence of $\Pi_m(t)$, obtained in stage 1, thus limits the number of modes to be kept in the stage 2 expansion depending on the evolution time $t$ and the desired accuracy. This will be illustrated for a simple example in Sec. 5. In Markovian semigroup limits this corresponds the well-known exponential decay that occurs between two 'quantum jumps'. The general influence of the cutoff $q_{max}$ on the quality of the time-evolution is, however, a nontrivial problem requiring further study.

Compare this with *TP approximations*. Under the TP constraint, discarding diagrams without further precautions will uproot the operator-sum structure of the expansion and thus violate CP. It is suggestive to directly relate the loss of (complete) positivity to an indication that not enough 'processes' were taken into account. However, the reason CP is lost is that one has discarded *partial contributions* to operationally well-defined processes which are CP. Nevertheless, once one has computed an approximate evolution map $\Pi$, one can check CP by calculating the associated Choi operator $\text{choi}(\Pi) = (\Pi \otimes \mathcal{I})|\mathbb{1}\rangle\langle\mathbb{1}|$ [Eq. (7)]. Violation of CP is then detected by finding a negative eigenvalue of this *operator*, but we are not aware of a reference that actually performs this check for a TP approximation.

At present there seems to be no guideline for systematically improving TP approximations 'towards CP' other than better approximating the exact result. Later on we will derive such a rule [Eq. (39)]. However, we must equally stress that because the TP constraint is less complicated, it allows much more advanced TP approximations which by their high accuracy may eventually be CP. This can however only be checked a posteriori and whether this outweighs the advantages of the CP approximations above seems difficult to say in general.

Finally, we stress that the above does not rule out the possibility of *CP-TP approximations* which *simultaneously* guarantee both properties. This requires schemes more sophisticated than the diagram selections considered so far. Building on Refs. [43,60,61], very recent work constructed systematic CP-TP approximations for a broad class of systems [62,63]. There it is noted that the CP constraint is trivially satisfied, but the TP constraint requires an ingenious truncation of environment correlation functions and the detailed proof of TP is nontrivial[19]. The difficulty of exactly enforcing TP in CP approximations was also noted in Refs. [44,70]. Approximated dynamics which is both CP and TP is furthermore interesting because it has a

---

[17]This is frequently used in quantum information where the full collection $\{\Pi_m\}$ is known as a quantum instrument and discarding terms corresponds to post-selection on measurement outcomes, see Ref. [78].

[18]This should be contrasted with Refs. [43,62,63] which introduces an expansion around the long-time limit: Whereas the leading order is exact in this limit, next-to-leading order corrections are only required to access times $t < \infty$ where non-Markovian effects are important.

[19]It can be shown that this truncation on the level of the Kraus operators used in Refs. [62,63] corresponds to a selection of Keldysh real-time diagrams for $\Pi$ which allow TP to be verified more easily using the rule (22), illustrating its usefulness. The novel feature of this diagram selection is that it entails a modification of the *pasting rule* (21b) which we kept fixed in our above considerations.

Figure 9: Hierarchies for propagators (a) on the double-branch and (b) on a single branch of the Keldysh contour. Each tier of the hierarchy corresponds to a *physical* process. The tier index (28) indicates the modes of the contracted lines in (a) and external uncontracted lines in (b). Horizontally cutting the $\Pi_m$ diagrams in (a) produces the corresponding $k_m$ diagrams in (b) and the converse is given by Eq. (21b). Cutting the *self-energy* diagrams $\Sigma_m$ in (a) produces the diagrams $\sigma_m$ in (b) when discarding one-branch reducible fragments. The nontrivial converse construction of $\Sigma_m[\sigma_0 \ldots \Sigma_m]$ is described in Sec. 4.3. Although this figure shows examples for bilinear coupling, all considerations also hold for multilinear coupling $V$.

Stinespring dilation [42]: Such dynamics can in principle always be expressed in terms of some effective joint unitary evolution $U'$ where some effective, initially uncorrelated environment $\rho^{E'}$ is traced out as in Eq. (1). Direct construction of the quantities $U'$ and $\rho^{E'}$ from $U$ and $\rho^E$, respectively, might be an interesting alternative route towards CP-TP approximations.

# 4 Hierarchies of self-consistent and kinetic equations

With so much room for approximations, it is of practical interest to have equivalent self-consistent and infinitesimal formulations of the measurement-conditioned evolution.

In Sec. 4.1 we show how to compute $\Pi_m$ directly from *memory kernels* $\Sigma_m$. This approach is closely related to well-developed techniques in statistical physics and is therefore discussed first. It avoids all use of the Keldysh operators $k_m$ (and Kraus operators) and instead implements stage 1 through the task of computing the self-energy *super*operators $\Sigma_m$.

In Sec. 4.2 we address the calculation of the Keldysh operators $k_m$ and by analogy introduce their *memory kernels* $\sigma_m$. This is possible since the $k_m$ are closely related to the $\Pi_m$ by the cutting rule. This instead implements stage 1 through the task of computing the self-energies $\sigma_m$ –operators on system and environment– and determines the Kraus operators of interest in quantum information theory.

The two approaches are graphically summarized in Fig. 9 and are equivalent, being related by $\Pi_m = \mathrm{Tr}_E : k_m : (\bullet \otimes \rho^E) : k_m^\dagger :$. In both approaches it remains to perform stage 2, the sum over modes $m$, collecting all CP contributions $\Pi = \sum_m \Pi_m$. The equations also share the same forward-feeding hierarchical structure which makes them of practical importance in view of the success of numerical methods [79] based on similar hierarchical formulations. However, the approaches are dual in that the relative difficulty of guaranteeing CP and TP are opposite. These two hierarchies allow us in Sec. 4.3 to derive an *infinitesimal* form of the operator-sum theorem by explicitly determining the functional relation between the families of memory kernels $\Sigma_m$ and $\sigma_m$.

## 4.1 Hierarchy for conditional evolutions $\Pi_m$

We start from the unraveling (21a) of the reduced evolution

$$\Pi(t) = \sum_m \Pi_m(t) \tag{26}$$

into conditional evolutions $\Pi_m$, each of which is a CP-TNI superoperator. Since $\Pi_m$ is the sum of all diagrams with $q$ fixed contractions involving modes $m = m_q \ldots m_1$ between opposite branches, one introduces corresponding *conditional self-energy* superoperators $\Sigma_m$ by restricting *all* contractions to be two-contour irreducible [Fig. 5(b)]. The full self-energy $\Sigma$ in the standard Dyson equation (18) –the sum over all two-branch irreducible diagrams– is obtained by summing over all modes $m$:

$$\Sigma(t,t') = \sum_m \Sigma_m(t,t'). \tag{27}$$

In the following we use simplified notation for the labels of the environment modes:

$$0 = \text{(no modes)}, \quad 1 = m_1, \quad 2 = m_2 m_1, \quad \ldots \tag{28}$$

As sketched in Fig. 9, the expansion (27) unravels the Dyson equation (18) into a hierarchy of equations (26) that is physically motivated: Each tier is labeled by the measurement outcome $m$, listing the environment modes that interacted with the system in the process $\Pi_m$. In the lowest tier of the hierarchy, $q = 0$, no modes interacted with system. We obtain an independent self-consistent equation

$$\Pi_0 = \pi + \pi * \Sigma_0 * \Pi_0, \tag{29a}$$

where $\pi$ denotes the free system propagator which reduces to $\pi = \mathcal{I}$ in the interaction picture. Importantly, this the only self-consistent step in the hierarchy. Its solution $\Pi_0 = \pi + \pi \sum_{l=1}^{\infty} (*\Sigma_0 * \pi)^l$ amounts to a renormalization of the free system propagator $\pi$, giving a CP-TNI superoperator, see Eq. (34) below[20]. Although $\Sigma_0$ involves all orders of the microscopic coupling $V$, it does not contain all diagrams of the standard $\Sigma$ in Eq. (18).

The solution of the zeroth tier (29a) goes into the following tiers with a simpler, forward-feeding structure:

$$\Pi_1 = \Pi_0 * \Sigma_1 * \Pi_0 \tag{29b}$$

$$\Pi_2 = \Pi_0 * \Sigma_2 * \Pi_0 + \Pi_0 * \Sigma_1 * \Pi_0 * \Sigma_1 * \Pi_0 \tag{29c}$$

$$\Pi_3 = \Pi_0 * \Sigma_3 * \Pi_0 + \Pi_0 * \Sigma_2 * \Pi_0 * \Sigma_1 * \Pi_0 + \Pi_0 * \Sigma_1 * \Pi_0 * \Sigma_1 * \Pi_0 * \Sigma_1 * \Pi_0 \tag{29d}$$
$$+ \Pi_0 * \Sigma_1 * \Pi_0 * \Sigma_2 * \Pi_0$$

$$\vdots$$

This enables an iterative solution of $\Pi_0$, $\Pi_1$, $\Pi_2$, …, given the conditional self-energies $\Sigma_0$, $\Sigma_1$, $\Sigma_2$, … and allows one to systematically implement approximation strategies discussed in Sec. 3.2. One checks that summing Eqs. (29) gives back the self-consistent Eq. (18).

In the hierarchy (29) it is implicitly understood that one symmetrizes[21] the mode indices appearing on the right hand side. Also note that on the right hand side individual contributions –or partial sums of them– do *not* correspond to a physical process: only their full sum makes

---

[20]This should be contrasted with the standard Dyson equation (18) which produces a solution $\Pi$ based on $\pi$, both of which are CP-TP, see App. C.

[21]For example, in equation (29c) the indices on the two $\Sigma_1$'s occurring in the second term on the right hand side of $\Pi_2 = \Pi_{m_1 m_2} = \Pi_{m_2 m_1}$ should be considered as independent variables: $\Sigma_1 * \Pi_0 * \Sigma_1 = \Sigma_{m_1} * \Pi_0 * \Sigma_{m_2} + (m_1 \leftrightarrow m_2)$.

up $\Pi_m$ which is a CP-TNI superoperator due to its operator-sum form (21b). Moreover, these terms consist of sequences of irreducible blocks $\Sigma_m$ and $\Pi_0$ and are thus *reducible*. This goes against the widespread intuition that contributions to irreducible kernels such as $\Sigma$ (and its components $\Sigma_m$) correspond to physical processes.

To further highlight how the physical processes $\Pi_m$ influence each other in *time*, we cast the self-consistent hierarchy (29) into an equivalent hierarchy of time-nonlocal kinetic equations for infinitesimal conditional evolutions: in the interaction picture

$$
\frac{d}{dt}
\begin{bmatrix}
\Pi_0 \\
\Pi_1 \\
\Pi_2 \\
\Pi_3 \\
\vdots
\end{bmatrix}
=
\begin{bmatrix}
\Sigma_0 & 0 & 0 & 0 & 0 & \cdots \\
\Sigma_1 & \Sigma_0 & 0 & 0 & 0 & \cdots \\
\Sigma_2 & \Sigma_1 & \Sigma_0 & 0 & 0 & \cdots \\
\Sigma_3 & \Sigma_2 & \Sigma_1 & \Sigma_0 & 0 & \cdots \\
\vdots & & & & \vdots
\end{bmatrix}
*
\begin{bmatrix}
\Pi_0 \\
\Pi_1 \\
\Pi_2 \\
\Pi_3 \\
\vdots
\end{bmatrix},
\tag{30}
$$

with the initial conditions $\Pi_0(0) = \mathcal{I}$ and $\Pi_m(0) = 0$ for $m \neq 0$. Each column of the matrix of self-energy superoperators sums to the standard self-energy $\Sigma = \sum_m \Sigma_m$, and by summing the column-entries of Eq. (30) to obtain $\Pi = \sum_m \Pi_m$, we reproduce the standard kinetic equation (3), $\frac{d}{dt}\Pi = \Sigma * \Pi$, in the interaction picture. Thus, the hierarchy presents an unraveling of the time-nonlocal kinetic equation (3). The lower-triangular form of the matrix ensures that the coupled kinetic equations feed forward and can be iteratively solved for $\Pi_0, \Pi_1, \ldots$, the formal solution being given by Eq. (29).

An advantage of the hierarchies over the standard (kinetic) equation Eq. (18) [Eq. (3)] is that both TP and CP have become explicit, even though the duality of their constraints persists [Sec. 2.4, 3.4]. Although the individual conditional self-energies do not preserve trace, i.e., $\mathrm{Tr}\,\Sigma_m \neq 0$ because the $\Pi_m$ are TNI, we can still see that $\Pi$ is TP: Summing the elements in each column of the matrix appearing in Eq. (30) and taking the trace gives $\mathrm{Tr}\,\Sigma = \sum_m \mathrm{Tr}\,\Sigma_m = 0$ because contributions from subsequent $\Sigma_m$ cancel pairwise by our earlier arguments[22].

As we know, CP is guaranteed for each $\Pi_m$ separately by Eq. (21a). In both hierarchies this is expressed by the forward-feeding structure which ensures that neglecting *complete higher tiers* (corresponding to larger number of modes $m_q \ldots m_1$) does not affect the lower tiers: if the latter are solved using the required finite set of *exact* $\Sigma_m$, the solutions are guaranteed to be CP. Thus, although there are no Kraus operators to be seen in Eq. (29)-(30), we can fully exploit the implications of the operator-sum theorem by unraveling the standard approach of statistical physics based on Eq. (3). At the same time, we can take advantage of the diagrammatic technique to compute the required (sub)set of conditional self-energies $\Sigma_m$ using rules which are very similar to the standard ones [App. D].

To set up more refined CP approximations one may consider additionally modifying the $\Sigma_m$ but doing so without uprooting CP is more complicated. In Sec. 3.3 we discussed a second approximation scheme which achieves this more easily by selecting one-branch diagrams for the Keldysh operators $k_m$. In the next section we set up a hierarchy tailored to this purpose.

## 4.2 Hierarchy for Keldysh operators $k_m$

Because the Keldysh operators $k_m$ are constructed from the $\Pi_m$ by the cutting rule [Sec. 3] they obey an analogous hierarchy of equations (32) illustrated in Fig. 9(b). In this case, we start from the unraveling (19) of the unitary evolution into its average $k_0$ and operators which

---

[22]TP is preserved through cancellation of pairs of diagrams appearing in different conditional evolutions: Eq. (24) implies that terms in $\Pi_{m_q \ldots m_1}$ cancel with those in 'adjacent' $\Pi_{m_q \pm 1 \ldots m_1}$. Since the argument leading to Eq. (24) also holds when removing $\Pi_0$ at the latest time, we find that contributions to the irreducible block $\Sigma_{m_q \ldots m_1}$ cancel with those in other blocks $\Sigma_{m_q \pm 1 \ldots m_1}$.

are explicitly normal-ordered:

$$U = k_0 + \sum_{m \neq 0} :k_m: \quad . \tag{31}$$

We now introduce $\sigma_m$ as the sum of all *one-branch* irreducible Keldysh half-diagrams representing $k_m$, i.e., irreducibility is understood with respect to the time-ordering on the upper branch of the Keldysh time-contour only. The Keldysh operators $k_m$ are then generated by the self-energy operators $\sigma_m$ through the following hierarchy[23] that inherits its structure from Eq. (29):

$$k_0 = u + u * \sigma_0 * k_0 \tag{32a}$$

$$k_1 = k_0 * \sigma_1 * k_0 \tag{32b}$$

$$k_2 = k_0 * \sigma_2 * k_0 + k_0 * \sigma_1 * k_0 * \sigma_1 * k_0 \tag{32c}$$

$$k_3 = k_0 * \sigma_3 * k_0 + \cdots \tag{32d}$$

$$\vdots$$

Here $u$ is the free evolution of system *and* environment which reduces to $u = \mathbb{1}_S \otimes \mathbb{1}_E$ in the interaction picture. As before, we symmetrize over the mode indices on the right hand side [Eq. (29)]. We note that the sum of Eqs. (32) results in the self-consistent equation

$$k = u + u * \sigma * k \tag{33}$$

for the operator $k(t) := \sum_m k_m(t)$ with $\sigma(t, t') := \sum_m \sigma_m(t, t')$. Importantly, this is *not* the self-consistent form of the Schrödinger equation due to the lack of normal-ordering relative to Eq. (31). Only *after explicitly normal-ordering* the hierarchy equations, the Schrödinger-Dyson equation, $U = u + u * (-iV)U$, with a *time-local* coupling $V$ is recovered, as it should. This nontrivial consistency check is worked out in App. B.2.

The zeroth tier Eq. (32a) is again an independent self-consistent equation. In this case, the self-energy $\sigma_0$ sums up all one-branch irreducible contractions without external fields and therefore acts trivially on the environment just as $k_0$. The solution $k_0 = u + u \sum_{l=1}^{\infty} (*\sigma_0 * u)^l$ of Eq. (32a) can then be related to the solution of Eq. (29a),

$$\Pi_0(t) = K_0^0(t) \bullet K_0^{0\dagger}(t), \tag{34}$$

noting that the zeroth operator [Eq. (15)] factorizes, $k_0 = K_0^0 \otimes \mathbb{1}_E = \langle U \rangle_E \otimes \mathbb{1}_E$, and that the free system evolution is $\pi = \mathrm{Tr}_E u(\bullet \otimes \rho^E) u^\dagger$. Although this separates the evolution into operators on the left and right, in general $K_0^0(t)$ is *not* generated by an effective nonhermitian Hamiltonian because $\sigma_0(t, t')$ is in general a time-nonlocal operator [cf. Eq. (42)].

Higher tiers determine the $k_m$ which act nontrivially on several modes $m$ of the environment. When one has solved the hierarchy for these operators one obtains the conditional propagators $\Pi_m$ by the pasting formula (21b). The Kraus operators $K_m^e(t) = \langle e | :k_m(t): \otimes \mathbb{1}_{E'} | 0 \rangle$ are determined by the solutions for $k_m(t)$ obtained from Eq. (32).

Taking the time-derivative of (32) we obtain a hierarchy of time-nonlocal kinetic equations for the Keldysh operators in the interaction picture,

$$\frac{d}{dt} \begin{bmatrix} k_0 \\ k_1 \\ k_2 \\ k_3 \\ \vdots \end{bmatrix} = \begin{bmatrix} \sigma_0 & 0 & 0 & 0 & 0 & \dots \\ \sigma_1 & \sigma_0 & 0 & 0 & 0 & \dots \\ \sigma_2 & \sigma_1 & \sigma_0 & 0 & 0 & \dots \\ \sigma_3 & \sigma_2 & \sigma_1 & \sigma_0 & 0 & \dots \\ \vdots & & & & & \vdots \end{bmatrix} * \begin{bmatrix} k_0 \\ k_1 \\ k_2 \\ k_3 \\ \vdots \end{bmatrix}, \tag{35}$$

---

[23]Refs. [43, 62, 63] derive a hierarchy for *Kraus* operators acting only on the system. Here we instead have a hierarchy of Keldysh operators which still act on the environment. Moreover, the hierarchy is expressed in terms of *self-energy* generators $\sigma_m$ which provides some advantages.

with the initial conditions $k_0(0) = \mathbb{1}$ and $k_m(0) = 0$ for $m \neq 0$. As before, these equations enable an iterative solution of $k_0$, $k_1$, $k_2$, etc., given the conditional self-energy operators $\sigma_0$, $\sigma_1$, $\sigma_2$, …. Importantly, this allows one to implement the second kind of approximation strategy discussed in Sec. 3.3. We note that summing the column-entries of Eq. (35) gives a single *time-nonlocal* evolution equation,

$$\frac{d}{dt}k = \sigma * k, \tag{36}$$

which is the differential formulation of Eq. (33), but *not* the Schrödinger equation due to the lack of normal-ordering in the quantity $k$.

Accounting for the normal-ordering, Eq. (35) [Eq. (32)] constitutes a time-nonlocal unraveling of the (Dyson) Schrödinger equation for system plus environment which has the advantage that it can be solved step-by-step. An advantage relative to Eq. (30) [Eq. (29)] is that the difficulty of guaranteeing CP and TP has been reversed. Indeed, in Eq. (32)-(35) we are free to formulate *any* approximation in terms of the *generators* $\sigma_m$ without *ever* compromising CP because they feed into the Keldysh operators $k_m$ constituting the CP operator-sum. Importantly, such $k_m$-approximations will not only *select* conditional evolutions $\Pi_m$ but also *modify* them [cf. Sec. 3.3].

Such a one-branch diagrammatic selection does not guarantee TP because the network of pairwise cancellations between different Keldysh operators [Eq. (24), Fig. 8] implies constraints that tie all $\sigma_m$ together. This makes TP harder to implement than the two-branch propagator hierarchy (30).

## 4.3   Operator-sum theorem and memory kernel of the quantum master equation

The formal solutions of the hierarchies in Sec. 4.1 and Sec. 4.2 are related by our key relation (21b), $\Pi_m = \mathrm{Tr}_E :k_m:(\bullet \otimes \rho^E):k_m^\dagger:$. The operator equations (35) [(32)] thus constitute an *exact* purification of the hierarchy of *super*operator equations (30) [(29)]. This implements the state-evolution correspondence [App. C] while explicitly separating out the normal-ordering procedure. The nontrivial reversal of the difficulty of imposing the constraints of CP and TP –despite the structural similarities of the two types of hierarchies– results from this duality and will now be traced *microscopically* to the difference in irreducibility and time-ordering on one and two time-branches: Whereas the self-energy superoperators $\Sigma_m$ were constructed from *two-branch* irreducible diagrams, the self-energy operators $\sigma_m$ are irreducible on a *single branch*, ignorant of time variables on the opposite branch[24].

To clarify this, we explicitly construct $\Sigma_m$ from the $\sigma_m$ by working out the nontrivial time-ordering constraints[25] in the Kraus *operator-sum theorem* (for finite time evolution) in order to transpose it to the level of the Nakajima-Zwanzig *memory-kernel* (for infinitesimal evolution). Here in particular, our diagrammatic approach shows its advantage: The functional form of the self-energies $\Sigma_m[\sigma_0 \dots \sigma_m]$ can be precisely determined starting from the prescription provided by the irreducible version of the pasting –or purification– equation (21b):

$$\Sigma_m[\sigma_0 \dots \sigma_m] = \mathrm{Tr}_E :k_m[\sigma_m \dots \sigma_0]: \big[\,\bullet \otimes \rho^E\,\big]:k_m^\dagger[\sigma_m^\dagger \dots \sigma_0^\dagger]:\Big|_{\mathrm{irred}}. \tag{37}$$

Inserting $k_m[\sigma_0 \dots \sigma_m]$ in terms of its generators, it remains to work out the *two-branch irreducibility* constraints on the one-branch time-integrations hidden inside $k_m$ and $k_m^\dagger$. We now

---

[24]This is a fundamental difference between the Liouville-von Neumann and Schrödinger equation which by our considerations ties in with the state-evolution correspondence.

[25]In special limits where the evolution is a dynamical semi-group this goes unnoticed: there one can easily apply the operator-sum theorem directly to $\Pi(t + dt) = \Pi(dt)\Pi(t)$, leading to the GKSL form of the quantum master equation [72]. Here we instead consider the more complicated *general* evolutions $\Pi(t)$ where this property fails.

$$\Sigma_0 = \text{(diagrams)} + \ldots + [\,\text{Swap}\,\updownarrow\,] \quad \text{(a)}$$

$$\Sigma_1 = \text{(diagrams)} + [\,\text{Swap}\,\updownarrow\,] \quad \text{(b)}$$

Figure 10: Conditional components $\Sigma_m$ of the memory kernel $\Sigma$ as explicit functional (39) of the Keldysh- and Kraus operator memory-kernels $\sigma_m$. The bold black lines indicate the full conditional propagator $k_0$ with no external vertex. (a) For $\Sigma_0$, two-branch irreducibility can only be ensured by alternating sequences of $\sigma_0$ $[\sigma_0^\dagger]$ on opposite branches, cf. Eq. (38). (b) For the remaining $\Sigma_m$, a dressed expansion can be set up in terms of the generators $\sigma_m$ following the rules (i)-(v) in the main text. This includes summation over all *definite two-branch time-orderings* of the self-energies $\sigma_m(\tau, \tau')$ [cf. Fig. 7(a)]. An explicit construction of $\Sigma_2$ is provided in App. F. The irreducible fragments $\Lambda_m$ obtained in step (iii) can be complemented by the generalization $\Lambda_0(t_+, t_-, t'_+, t'_-)$ of Eq. (39a) to a four-time object, see rule (iv).

show that this amounts to a 'dressed' real-time expansion on the Keldysh contour in terms of effective vertices $\sigma_0, \sigma_1, \ldots$ with an intricate internal time-ordering structure.

In Fig. 10(a) we sketch the case for $\Sigma_0$ where the two-branch irreducibility can be achieved in only two ways using one-branch irreducible building blocks. Either a generator $\sigma_0$ (green) spans over one whole branch (first diagram); then the time-integrations on the other branch remain unrestricted and the full one-branch propagator $k_0$ appears (bold line). Otherwise, the two-branch irreducibility can only be maintained by an alternating sequence of generators $\sigma_0$ on the upper and $\sigma_0^\dagger$ on the lower branch with temporal overlaps, filled up with $k_0^\dagger$ resp. $k_0$ propagations in between (further diagrams). We thus see that the basic blocks of the memory kernel on two branches are *four-time* superoperators[26] describing irreducible time-evolution from time $t'_+$ to $t_+$ on the upper branch and reducible evolution from $t'_-$ to $t_-$ on the lower branch,

$$\Lambda_0^+(t_+, t_-, t'_+, t'_-) = \sigma_0(t_+, t'_+) \bullet k_0^\dagger(t_-, t'_-), \tag{38a}$$

with $t_+ \geq t_- \geq t'_- \geq t'_+$ and conversely

$$\Lambda_0^-(t_+, t_-, t'_+, t'_-) = k_0(t_+, t'_+) \bullet \sigma_0^\dagger(t_-, t'_-), \tag{38b}$$

with $t_- \geq t_+ > t'_+ \geq t'_-$. Two-branch irreducibility is enforced by letting the one-branch irreducible generator $\sigma_0$ ($\sigma_0^\dagger$) start earlier and end later than their reducible counterparts $k_0^\dagger$ ($k_0$) on the opposite branch. In terms of these auxiliary objects the functional form of $\Sigma_0$ can be written down explicitly:

$$\Sigma_0(t, t') = \sum_{n=1}^{\infty} \sum_{p=\pm} \left[ \Lambda_0^p \star \Lambda_0^{-p} \star \ldots \star \Lambda_0^{(-p)^{n-1}} \right](t, t, t', t'). \tag{39a}$$

Here, the two-branch convolution

$$\left[ \Lambda_0^+ \star \Lambda_0^- \right](t_+, t_-, t'_+, t'_-) := \int_{t'_+}^{t_+} d\tau_+ \int_{t'_-}^{t_-} d\tau_- \, \Lambda_0^+(t_+, t_-, \tau_+, \tau_-) \Lambda_0^-(\tau_+, \tau_-, t'_+, t'_-)$$

---

[26]Similar objects are considered in Refs. [43, 62, 63].

accounts for independent one-branch convolutions imposing the two-branch time-ordering constraints of Eq. (38).

Once $\Sigma_0$ has been constructed in this way, the remaining tiers of the self-energy $\Sigma_m$ can be obtained from a dressed expansion governed by the following diagrammatic rules illustrated in Fig. 10(b). We start from a Keldysh contour where each branch, running from the external times $t'$ to $t$, is dressed to the Keldysh operator $k_0$ [$k_0^\dagger$]. To obtain $\Sigma_{m_q \ldots m_1}(t, t')$ for fixed modes $m_q \ldots m_1$ perform the following steps:

(i) Distribute in all possible ways the $q$ external vertices acting on the fixed set of modes $m_q \ldots m_1$ among self-energy blocks $\sigma_\mu$ [$\sigma_\mu^\dagger$] with $\mu \neq 0$ on the upper [lower] branch.

(ii) Integrate over all internal time-arguments of these blocks, but *restrict* them to *definite two-branch time-orderings*, see the first three contributions in Fig. 10(b). Let the earliest (latest) self-energy block start (end) at $t'$ ($t$). Note that a single block may extend over a whole branch. Sum over the *finite* number of possibilities of such orderings.

(iii) Contract the external vertices in all possible ways to obtain *two-branch irreducible fragments* $\Lambda_\mu(t_+, t_-, t'_+, t'_-)$ as shown for the first three contributions in Fig. 10(b). Analogous to Eq. (38), the four time-arguments indicate propagation times on the upper and lower contour respectively .

(iv) Insert the four-time generalization of Eq. (39a), $\Lambda_0(t_+, t_-, t'_+, t'_-)$, at all positions where the two-branch irreducibility is not yet ensured. Such cases only occur for tiers with more than one mode ($q \geq 2$), see App. F. This explicitly enforces two-branch irreducibility.

(v) Add all irreducible extensions of the constructed diagrams by attaching $\Lambda_0$ on the left or on the right or on both sides as shown in the last contributions of Fig. 10(b).

Due to this generic structure, the two-branch irreducible fragments $\Lambda_\mu[\sigma_0 \ldots \sigma_m]$ form yet another hierarchy for the conditional self-energy superoperators,

$$\Sigma_1(t, t') = \ \big[(\Delta + \Lambda_0) \star \Lambda_1 \star (\Delta + \Lambda_0)\big](t, t, t', t') \tag{39b}$$

$$\Sigma_2(t, t') = \ \big[(\Delta + \Lambda_0) \star \Lambda_2 \star (\Delta + \Lambda_0)\big](t, t, t', t')$$
$$+ \big[(\Delta + \Lambda_0) \star \Lambda_1 \star \Lambda_0 \star \Lambda_1 \star (\Delta + \Lambda_0)\big](t, t, t', t') \tag{39c}$$
$$\vdots$$

where the two-branch convolution takes care of the intricate time-ordering required for CP. Here $\Delta(t_+, t_-, t'_+, t'_-) = \delta(t_+ - t'_+) \delta(t_- - t'_-)$ is a shorthand for 'no insertion'. Except for the $\Delta$'s, $\Sigma_m$ inherits a structure similar to that of the hierarchy (29) for $\Pi_m$. Also here symmetrization over the mode labels is understood implicitly.

The above diagrammatic rules allow different terms of the functional to be explicitly written down. In App. F we illustrate this for a nontrivial example in second order of the dressed expansion. Importantly, it is again *only* the zeroth tier $\Sigma_0$ that requires summation over an *infinite* set of diagram contributions via the relation (39a). The remaining contributions –dressed up by the so obtained $\Sigma_0$– represent a *finite* number of terms to evaluate. As of course expected for this general problem, evaluating these terms is challenging. In the limits where the GKSL form applies the arising complications disappear and the Lindblad jump operators automatically emerge from the conditional self-energies $\sigma_m(t, t')$ of the Keldysh operators with external fields ($m \neq 0$) as the example in Sec. 5 illustrates.

*Approximations*. The functional form (39) precisely encodes the nontrivial structure of the memory-kernel of the Nakajima-Zwanzig quantum master equation imposed by the Kraus operator-sum theorem. This means that *whatever* (approximated) $\sigma_m$ one uses, a memory kernel $\Sigma_m$ (or $\Sigma = \sum_m \Sigma_m$) with the functional form (39) will generate a CP evolution $\Pi_m$ ($\Pi$) through the kinetic equation (30) [Eq. (3)]. It is *this* structure of the kernel –enforced by Eq. (37)– that guarantees that the solution can be expressed as $\Pi_m = \mathrm{Tr}_E : k_m : (\bullet \otimes \rho^E) : k_m^\dagger :$.

The practical importance of the nontrivial functional (39) is that it establishes a microscopic framework for approximations that go beyond the restrictive limits of the GKSL approach without giving up CP. Knowing the explicit structure (39) of the self-energy is also a prerequisite for setting up functional renormalization-group (RG) approaches. It may thus provide a starting point for connecting the *exact* hierarchies of RG equations formulated on one [80] and two [34, 41] branches of the Keldysh contour while explicitly preserving the CP structure of the density operator evolution. This would allow one to assess the implications for CP of truncation schemes for such RG hierarchies. Interestingly, some RG approaches [34, 37, 41] also entail a two-stage approach similar in spirit to ours but based on an RG flow starting from the Markovian $T = \infty$ limit [24, 37] that we study next in Sec. 5.

We stress that the functional (39) is also relevant for less ambitious efforts. Consider, for example, a TP-approximation based on a selection of diagrams [26–31] that breaks CP [Sec. 3.5]. Then the functional (39) allows one to systematically identify which groups of diagrams are missing, what their time-ordering structure is, and provides a guide for adding *some* –but not all– terms to *improve* towards[27] a complete operator-sum form. This amounts to partially 'completing the square' of the *solution* $\Pi$, while working directly on the level of the self-energy of the *kinetic equation*.

We note that the physical structure of the memory kernel $\Sigma(t, t')$ is often identified with its role as a response function which is tied to definite analytic properties [34, 41] in the Laplace-frequency representation $\Sigma(E) = \int_0^\infty ds e^{-iEs} \Sigma(s)$ in time-translationally invariant cases where $\Sigma(t, t') = \Sigma(t - t')$. Moreover, transition frequencies of coherent oscillations and decay rates in the dynamics $\Pi(t)$ are encoded in the poles and branch points of $\Sigma(E)$. How this simple analytic structure can be extracted from the *real-time* Kraus operators is unclear. Indeed, the Laplace-transformed Kraus operators have an extremely complex explicit structure [62, 63]. However, it is important to note that the analytic structure does *not* exhaust the physical content of the memory kernel because it is actually more than a response function: It is also a superoperator and it should therefore satisfy additional *algebraic* requirements that ensure the dynamics it generates is CP. Our result (39) demonstrates that the memory kernel indeed has such a nontrivial additional structure in which time-integration constraints are coupled to the algebraic building blocks. Without this structure the memory kernel may well have the above mentioned analytic structure but *still* generate, instead of a CP map, a PP map without a physical / operational meaning, as explained in Sec. 2.1. It is unclear how the algebraic requirements can be extracted from the frequency dependence of the memory kernel $\Sigma(E)$.

# 5 Simple example: Fermionic mode coupled to a hot fermion bath

In this final section we illustrate the developed ideas on a simple example for which all calculations can be explicitly followed, both on the level of the Keldysh operators $k_m$ and the conditional propagators $\Pi_m$. The aim is to illustrate *how* CP is guaranteed explicitly by the dual hierarchies when systematically constructing the self-energies and then solving each of their tiers. Although other methods also solve this model [see Eq. (52)], these do not show how CP is guaranteed in the *general* case of ultimate interest. Throughout this section, we will work in the Schrödinger picture.

It suffices to consider the model (4) of a fermion mode coupled to a continuous reservoir of noninteracting fermions held at infinite temperature ($T = \infty$). As we noted [Eq. (6)], in this example each mode of the environment is labeled by a particle-hole index and a continuous energy, $m_i = \eta_i \omega_i$. The combined $T = \infty$ and wideband limit[28] leads to simplifications

---

[27]The improvement can be monitored via the Choi operator (7) associated with $\Pi$ [Sec. 3.5, App. C].

[28]At finite temperatures, cancellations between Keldysh diagrams can also be exploited to solve this problem,

$$
k_0 = \text{——} + \overset{\sigma_0}{\text{⎘}} + \text{⎘⎘} + \ldots = \text{——} \quad (a)
$$

$$
k_1 = \overset{\sigma_1}{\text{⊥}} \quad\longleftrightarrow\quad \int_{\omega_1} \Pi_1 = \overset{\Sigma_1}{\text{⊤}} \quad (b)
$$

$$
k_2 = \overset{\sigma_2 = 0}{\text{⊤⊥}} \quad\longleftrightarrow\quad \int_{\omega_1\omega_2} \Pi_2 = \overset{\Sigma_2 = 0}{\text{⊓}} \quad (c)
$$

$$\vdots$$

Figure 11: Hierarchy of the Keldysh-operators (left) and conditional propagators (right) for the model (4). The combined infinite-temperature and wideband limit forces all contractions to be instantaneous in time (drawn as vertical) suppressing all vertex correction diagrams. In combination with the bilinearity of the coupling $V$ this causes all self-energies $\sigma_m$ and $\Sigma_m$ of tier $m \geq 2$ to vanish identically, but only after integrating over the environment mode energies $\omega_i$, thereby partially performing stage 2.

whenever we contract environment modes *and* sum over the mode energy,

$$
\int_\omega \frac{\Gamma}{2\pi} \langle b_{\eta'\omega'} e^{-i\eta\omega(\tau-\tau')} b_{\eta\omega} \rangle = \frac{\Gamma}{4}\delta(\tau-\tau')\delta_{\eta',-\eta}. \tag{40}
$$

This limit therefore causes strictly Markovian semi-group dynamics and renders the model solvable. Here and below, all $\delta$-functions are adapted to working with time-convolutions: they are defined such that $\int_0^\tau d\tau'\delta(\tau-\tau') = 1$ instead of $1/2$. We note upfront that the $T = \infty$ and wideband limit needs to be taken *twice*: once, when we compute the conditional propagator $\Pi_m$ in stage 1 by summing one-branch-diagrams to obtain $k_m$ which is then pasted together with $k_m^\dagger$ using Eq. (21b); we take the limit a second time in stage 2 where we integrate over the environment energy $\omega$ contained in the mode index $m = \eta\omega$. Importantly, the pasting result (21b) of stage 1 *cannot* be simplified without this stage 2 energy-integration. As a result, the stage 1-2 distinction is easily blurred when simplifying expressions. We will first illustrate this for $k_m$ in Sec. 5.1. This becomes a more prominent issue when computing $\Pi_m$ directly –without the use of Keldysh operators– which we illustrate in Sec. 5.2.

## 5.1 One-branch hierarchy for $k_m$

Starting with the lowest tier of the hierarchy Eq. (32), we first have to compute the generators for the Keldysh operators. Only the single bubble-diagram depicted in Fig. 11(a) contributes which translates to

$$
\sigma_0(\tau, \tau') = \left\langle e^{iH^0\tau}(-iV)e^{-iH^0(\tau-\tau')}(-iV)e^{-iH^0\tau'} \right\rangle_{\mathrm{E}} \otimes e^{-iH^{\mathrm{E}}(\tau-\tau')}
$$

$$
= -\frac{\Gamma}{4}\delta(\tau-\tau')\sum_\eta d_\eta d_{-\eta} \otimes e^{-iH^{\mathrm{E}}(\tau-\tau')} = -\frac{\Gamma}{4}\delta(\tau-\tau')\mathbb{1}_{\mathrm{S}} \otimes \mathbb{1}_{\mathrm{E}}. \tag{41}
$$

We next construct the Keldysh operator by iterating the self-consistent equation (32a):

$$
k_0(t) = e^{-iH^0 t} + \int_0^t d\tau \int_0^\tau d\tau' e^{-iH^0(t-\tau)}\sigma_0(\tau,\tau')k_0(\tau')
$$

$$
= e^{-iH^0 t} + \int_0^t d\tau\, e^{-iH^0(t-\tau)}\left[-\frac{\Gamma}{4}\mathbb{1}_{\mathrm{S}} \otimes \mathbb{1}_{\mathrm{E}}\right]k_0(\tau) = e^{-iH^0_\infty t}. \tag{42}
$$

---

see Sec. 4.2 of Ref. [77].

In contrast to the general case [Eq. (34) ff.], the zeroth tier evolution is generated by a non-hermitian Hamiltonian for system and environment, $H_\infty^0 := H^0 - i\frac{\Gamma}{4}\mathbb{1}_S \otimes \mathbb{1}_E$, because $\sigma_0$ is time-local in this model.

As sketched in Fig. 11(b), the next tier self-energy operators are

$$\sigma_{\eta_1\omega_1}(\tau,\tau') = i\delta(\tau-\tau')\sqrt{\frac{\Gamma}{2\pi}}\eta_1 d_{\eta_1}^\dagger b_{\eta_1\omega_1}, \qquad (43)$$

where we explicitly write the mode index $1 = m_1 = (\eta_1\omega_1)$. Using this we can then evaluate Eq. (32b) for the Keldysh operator without the need for iteration:

$$\begin{aligned}
k_{\eta_1\omega_1}(t) &= \int_0^t d\tau \int_0^\tau d\tau' k_0(t-\tau)\sigma_{\eta_1\omega_1}(\tau,\tau')k_0(\tau') \\
&= i\eta_1\sqrt{\frac{\Gamma}{2\pi}}\int_0^t d\tau e^{i\eta_1(\omega_1-\varepsilon)\tau} e^{-iH_\infty^0 t} d_{\eta_1}^\dagger b_{\eta_1\omega_1}.
\end{aligned} \qquad (44)$$

For higher tiers $q \geq 2$, the self-energies vanish $\sigma_{m_q\ldots m_1} = 0$ because the wideband and $T = \infty$ limit suppresses all *one-branch* vertex corrections. Thus, only repeated convolutions of $k_1$ remain for the $q$-th tier equation and we obtain

$$k_{\eta_q\omega_q\ldots\eta_1\omega_1}(t) = \frac{1}{q!}\prod_{j=1}^q \left(i\eta_j\sqrt{\frac{\Gamma}{2\pi}}\int_0^t d\tau_j e^{i\eta_j(\omega_j-\varepsilon)\tau_j}\right) e^{-iH_\infty^0 t} d_{\eta_q}^\dagger b_{\eta_q\omega_q}\ldots d_{\eta_1}^\dagger b_{\eta_1\omega_1}, \qquad (45)$$

with implicit symmetrization over the mode-indices. The time-ordering on the integrals has been lifted with the prefactor $1/q!$ accounting for the corresponding double-counting. We see that the decay rate $\Gamma$ inside the *exponential* (42) appearing in Eq. (45) is generated initially when solving the lowest hierarchy-tier for $k_0(t)$. Through the hierarchy it sets the timescale for all higher $k_m(t)$ *before* we have integrated out the meter M by performing the sum over $m$. We stress that to further simplify the results for $\Pi_m$ in stage 1 we *must* partially perform stage 2, $\sum_m \Pi_m$, in order to exploit the wideband and $T = \infty$ limit for a *second* time: only when integrating $\Pi_m = \text{Tr}_E :k_m:(\bullet \otimes \rho^E):k_m^\dagger:$ over the energies $\omega_i$ of the modes $m_i$ (leaving the $\eta_i$ untouched) can the time-integrations in Eq. (45) be reduced to further $\delta$-functions in time.

## 5.2 Two-branch hierarchy for $\Pi_m$

First, the zeroth tier conditional propagator is obtained directly by pasting together the Keldysh operators (42):

$$\Pi_0(t) = \text{Tr}_E k_0[\bullet \otimes \rho^E]k_0^\dagger = e^{-iH_\infty t} \bullet e^{-iH_\infty^\dagger t} := K_0 \bullet K_0^\dagger, \qquad H_\infty := \varepsilon d^\dagger d - i\frac{\Gamma}{4}\mathbb{1}_S. \qquad (46)$$

Alternatively, one computes $\Sigma_0$ from (39a), where due to time-locality only the first type of diagram in Fig. 10(a) contributes:

$$\Sigma_0(\tau,\tau') := \sigma_0(\tau,\tau')\bullet + \bullet \sigma_0^\dagger(\tau,\tau') = -\frac{\Gamma}{2}\delta(\tau-\tau')\mathbb{1}_S \otimes \mathbb{1}_E. \qquad (47)$$

For the next tier, we can again proceed in two equivalent ways. The direct approach inserts Eq. (44) into the pasting formula $\Pi_1 = \text{Tr}_E :k_1:(\bullet \otimes \rho^E):k_1^\dagger:$ to obtain Eq. (49) below. Alternatively, one can bypass computing $k_1$ altogether by computing $\Sigma_1$ and then generating $\Pi_1$ through the two-branch hierarchy. Using our general functional (39b) to compute

$$\int_{\omega_1} \Sigma_{\eta_1\omega_1} = \int_{\omega_1} \diagdown + \diagup = \big| + \big|$$

Figure 12: Energy-integrated functional $\Sigma_1[\sigma_0, \sigma_1]$, cf. Fig. 10(b). Due to the energy integration, the two-branch convolution reduces to a $\delta$-function in time.

$\Sigma_1[\sigma_0, \sigma_1]$, only the single pair of diagrams shown in Fig. 12 contributes. Its construction from $k_0$ [Eq. (42)] and $\sigma_1$ [Eq. (43)] is straightforward with the two-branch convolution:

$$\int_{\omega_1} \Sigma_{\eta_1\omega_1}(t, t') = \int_{\omega_1} \left[ (\sigma_{\eta_1\omega_1} \bullet k_0^\dagger) \star (k_0 \bullet \sigma_{\eta_1\omega_1}^\dagger) + (k_0 \bullet \sigma_{\eta_1\omega_1}^\dagger) \star (\sigma_{\eta_1\omega_1} \bullet k_0^\dagger) \right](t, t, t', t')$$

$$= \frac{\Gamma}{2} \delta(t - t') d_{\eta_1}^\dagger \bullet d_{\eta_1}. \tag{48}$$

As pointed out above, we are now forced to integrate over the mode energies to exploit the wideband and $T = \infty$ limit a *second* time. This causes the general functional (39) to vastly simplify due to the time-locality of the contractions between the *opposite* time-branches. Equivalent manipulations are also required in the evaluation of the direct approach.

Analogous to Eq. (44), we can then evaluate the first tier equation (29b) without iteration for the conditional propagator:

$$\int_{\omega_1} \Pi_{\eta_1\omega_1}(t) = \Pi_0 * \int_{\omega_1} \Sigma_{\eta_1\omega_1} * \Pi_0(t) = \frac{\Gamma}{2} t e^{-iH_\infty t} d_{\eta_1}^\dagger \bullet d_{\eta_1} e^{iH_\infty t}. \tag{49}$$

Importantly, the time convolution produced the linear factor $t$ with time-scale $\frac{\Gamma}{2} \sim \Sigma_1$.

All higher self-energies identically vanish only when *integrated* over mode energies $\int_{\omega_q \dots \omega_1} \Sigma_{m_q \dots m_1} \equiv 0$ for $q \geq 2$ where the wideband and $T = \infty$ limits suppress vertex corrections generated by contractions between *two branches*. For the $q$-th tier equation integrated over the energies only repeated convolutions of $\Sigma_1$ remain,

$$\Pi_{\eta_q \dots \eta_1}(t) := \int_{\omega_q \dots \omega_1} \Pi_{\eta_q \omega_q \dots \eta_1 \omega_1}(t) = \Pi_0 * \int_{\omega_q} \Sigma_{\eta_q \omega_q} * \Pi_0 * \dots * \int_{\omega_1} \Sigma_{\eta_1 \omega_1} * \Pi_0(t)$$

$$= \int_{\omega_q} \Pi_{\eta_q \omega_q} * \dots * \int_{\omega_1} \Pi_{\eta_1 \omega_1}(t) = K_{\eta_q \dots \eta_1}(t) \bullet K_{\eta_q \dots \eta_1}^\dagger(t), \tag{50}$$

where we used $\Pi_0 * \Sigma_1 * \Pi_0 = \Pi_1$ and inserted Eq. (49). This agrees with the direct approach which pastes $k_m$ [Eq. (45)] to obtain $\Pi_m$. For each set of modes *integrated* over their frequencies, we get a *single* Kraus operator[29]:

$$K_{\eta_q \dots \eta_1}(t) = \sqrt{\frac{(\frac{\Gamma}{2}t)^q}{q!}} e^{(-i\varepsilon d^\dagger d - \frac{1}{4}\Gamma)t} d_{\eta_q}^\dagger \dots d_{\eta_1}^\dagger. \tag{51}$$

The *prefactor* $(\frac{\Gamma}{2}t)^q/q!$ is generated *only* in Eq. (50) where we integrate over the mode-energies of the inter-branch contractions, i.e., we partially evaluated stage 2. Doing so results in the

---

[29]It should be noted that the conditional propagator (50) is symmetric in the mode-indices as required, whereas the Kraus operators (51) themselves are not: Possible signs from permuting the fermionic fields $d_{\eta_i}$ in the Kraus operators exactly cancel when plugged into the propagator.

$\delta$-functions (second wideband limit), and the time-ordered integral reduces to $t^q/q!$. As we noted at Eq. (45), this *cannot* be seen in $k_m$ or $\Pi_m$ obtained in stage 1.

We complete stage 2 by summing over the particle-hole indices $\eta_i$,

$$\sum_{q=0}^{\infty} \sum_{\eta_q \cdots \eta_1} \Pi_{\eta_q \cdots \eta_1} = \sum_{q=0}^{\infty} \sum_{\eta_q \cdots \eta_1} \frac{(\frac{\Gamma}{2}t)^q}{q!} e^{-\frac{\Gamma}{2}t} e^{-i\varepsilon t d^\dagger d} d_{\eta_q}^\dagger \cdots d_{\eta_1}^\dagger \bullet d_{\eta_1} \cdots d_{\eta_q} e^{+i\varepsilon t d^\dagger d} = e^{\mathcal{L}t}, \quad (52)$$

and find the GKSL generator $\mathcal{L} = -i[\varepsilon d^\dagger d, \bullet] + \frac{\Gamma}{2} \sum_\eta \left( -\frac{1}{2}[d_\eta d_\eta^\dagger, \bullet] + d_\eta^\dagger \bullet d_\eta \right)$. Although we could have derived [48] a GKSL master equation $\partial_t \rho(t) = \mathcal{L}\rho(t)$ –which is exact for this model– to more easily obtain the result (52), all steps and physical arguments that we discussed can also be applied for *general* evolutions –for which *no* GKSL equation can be derived. Clearly, in the latter case the evaluation steps will be technically more difficult.

*Discussion*. The two distinct occurrences of the coupling constant $\Gamma$ in the simple results for the *time-dependence* of the Keldysh operator (45) and the Kraus operator (51) illustrate the two fundamental stages of the real-time expansion. As we anticipated in Sec. 3.3, the physically motivated hierarchy index $m = \eta\omega$ (measurement outcomes) 'counts the powers' in stage 2 of the expansion. This yields the series in powers $q$ of *the prefactor* $\frac{\Gamma}{2}$ in Eq. (52) which is CP order-by-order. We now explicitly see that the $q$-expansion does not correspond to a naive short-time expansion [Sec. 3.5]: Each term in Eq. (52) with fixed $q$ –obtained by summing up all intra-branch contractions [Eq. (42)]– decays expontially with rate $\frac{\Gamma}{2}$. The contribution of this $q$-th order process is maximal for times $t = q(\frac{\Gamma}{2})^{-1}$. Thus, the cutoff $\Gamma$ in the exponential, generated in stage 1, controls how many terms in stage 2 should be kept for a good approximate result. The quality of a CP approximation to Eq. (52) in which we ignore *physical processes* involving at most $q_{max}$ environment modes is quantified by a bounded TP error [cf. Sec. 3.5],

$$0 \leq 1 - \sum_{q=0}^{q_{max}} \frac{(\frac{\Gamma}{2}t)^q}{q!} e^{-\frac{\Gamma}{2}t} \leq 1. \quad (53)$$

Due to the operational nature of this approximation, the error (53) monotonically decreases as $q_{max}$ is increased. The required value of $q_{max}$ for the TP error to be negligible depends on the time scale of interest. More specifically, consider an initially occupied level, $\rho(0) = |+\rangle\langle+|$, for which the exact occupation decays as $\langle+|\rho(t)|+\rangle = \frac{1}{2}(1 + e^{-\Gamma t})$. The CP approximation truncated at even $q_{max} = 2k_{max}$ gives $\langle+|\rho(t)|+\rangle = e^{-\frac{\Gamma}{2}t} \sum_{k=0}^{k_{max}} (\frac{\Gamma}{2}t)^{2k}/(2k)!$. Already for a finite $q_{max} = 8$, this is indistinguishable from the exact result well up to the point where the stationary value $1/2$ is reached. The nontrivial part of the evolution has thus already been captured and the *only* effect of further increasing $q_{max}$ is to maintain the constant stationary value up to larger times. This illustrates that in CP approximations the loss of TP need not be an insurmountable problem and one can still benefit from a well-defined error which does not hide cancellations. However, it also shows that for CP approximations one cannot *directly* work in the stationary limit $t \to \infty$. If this is of primary interest, then TP approximation are more advantageous.

Finally, we note that the above illustrates how easily limiting procedures –even in an exactly solvable problem– can hide the CP structure (21) of the real-time expansion. We stress that even though the energy integrations –partially accomplishing stage 2– were necessary to analytically solve the problem, they were *never* needed to reveal CP: In our formulation at each step the conditional propagators have the operator-sum form. We also note that Eq. (50) written *without* energy integrations requires the higher self energies $\Sigma_m$ to recover the explicit operator sum $\Pi_m = \text{Tr}_E :k_m:(\bullet \otimes \rho^E):k_m^\dagger:$. Then, even in this simple model, vertex corrections are relevant and the $\Sigma_m$ are *not* zero for $m \neq 0, 1$.

# 6   Summary and discussion

In this paper we studied the dynamics of the reduced density operator $\rho(t) = \Pi(t)\rho(0)$ by combing the operator-sum representation –ubiquitous in quantum-information theory and tied to entanglement– with the Keldysh real-time expansion of statistical field theory tailored to a microscopic description of continuous environment models.

*Measurement-conditioned evolution*. We decomposed the reduced evolution $\Pi$ into elementary conditional evolutions $\Pi_m$ which are completely positive *because* they correspond to possible measurement outcomes $m$ in a quantum circuit [Fig. 3]. By insisting on such a physical / operational formulation, we recovered the standard real-time expansion in a crucially reorganized two-stage form, revealing it actually consists of two intertwined expansions [Eq. (21)]. Based only on fundamental principles of measurements, this extends the idea of an expansion in alternating 'life-time broadened evolution' and 'quantum-jumps' as known from GKSL equations to general CP-TP dynamics (1). This is a crucial step to broaden the limited scope of open-system problems that are addressable when strictly adhering to the fundamental CP principle of quantum information.

*Cutting diagrams – unraveling the Schrödinger equation*. By cutting the $\Pi_m$ superoperator diagrams on the Keldysh double-time contour into halves we obtained what we called *Keldysh operators* $k_m$ on a single time branch. These 'lift' the *Kraus operators* $K_m^e(t) = \langle e | : k_m(t) : \otimes \mathbb{1} | 0 \rangle$ to act on both system *and* environment, encoding their *time-evolution* [Eq. (14)] *and* the Wick normal-ordering. From the obtained diagrammatic expansion we inferred time-nonlocal memory kernels for these quantities and derived a hierarchy of self-consistent evolution equations for the Keldysh operators [Eq. (32)] . This hierarchy generates non-Hamiltonian dynamics on one time branch.

The corresponding hierarchy [Eq. (35)] of *kinetic* equations for Keldysh operators $k_m$ constitutes a time-nonlocal unraveling of the Schrödinger equation conditioned on the measurements $m$. The advantage over the original Schrödinger equation is that these one-branch kinetic equations already explicitly encode the operator-sum decomposition that guarantees CP *before* having integrated out the environment completely.

*Pasting diagrams – unraveling the quantum master equation*. By pasting the obtained Keldysh / Kraus operator diagrams back together –reconnecting the time branches– we obtained an equivalent self-consistent hierarchy [Eq. (29)] for conditional evolutions $\Pi_m$. This expresses the operator-sum theorem on the Keldysh real-time contour of statistical field theory.

The corresponding hierarchy [Eq. (30)] of *kinetic* equations for conditional evolutions $\Pi_m$ constitutes a physical / operational unraveling of the general (Nakajima-Zwanzig) quantum master equation. We expressed its two-time-branch memory kernel superoperator explicitly in terms of the one-time-branch memory kernels of the Keldysh operators, the *building blocks of the Kraus operators* [Eq. (39)]. Irrespective of which approximations to the latter are inserted, a Nakajima-Zwanzig memory-kernel with this functional *form* generates solutions which are guaranteed to be CP. This algebraic constraint on the memory-kernel –operationally derived in terms of measurements– complements the well-known analytic structure of its frequency representation –derived from its role as a causal response function.

The tiers of the hierarchies for $k_m$ and $\Pi_m$ are labeled by the modes which have interacted with the system as determined by a *possible measurement* on the environment. This operational nature of the hierarchies guarantees CP tier-by-tier. We have illustrated how this hierarchy can be truncated based on the dissipative time-scale generated at an earlier stage [Eq. (51)].

*The role of time*. A recurring theme has been the complementarity between the approaches from quantum-information and statistical field theory that we combined. This was not entirely unexpected given the celebrated Choi-Jamiołkowski correspondence between quantum evolutions and bipartite quantum states [74–76]. However, we showed how this entails a duality in

approximating reduced time-evolutions –strictly fixing CP tends to uproot TP and *vice versa*– and our two types of self-consistent and kinetic equations explicitly encode this. There is thus a *deeply physical / operational* reason why the two approaches to open systems are so different.

By descending to the level of the microscopic coupling $V(t)$ –instead of staying on the formal level of the joint unitary $U(t) = Te^{-i \int_0^t d\tau V(\tau)}$– we exposed the crucial role of *time* in this CP-TP duality. More precisely, we found the duality is related to the intricate ordering of time on the Keldysh contour: On one branch time orders operators, whereas it orders superoperators on two branches. The operational (input-output) approach of quantum information theory ignores time and hides the one-branch time-ordering inherited from the unitary $U$ *inside* the Kraus operators. This trivializes the CP structure through the operator-sum theorem, but makes TP seem a nontrivial 'global' constraint in practice. In contrast, the statistical physics approach, formulated on the two branches of the Keldysh contour, trivializes the TP structure because the trace connects the two branches, but thereby promotes CP to a nontrivial 'global' constraint. Importantly, on the *microscopic* level of the coupling $V$ we found that these two approaches resolve each other's enigmas: The nontrivial TP constraint for Kraus / Keldysh operators living on one time branch is achieved by pairwise cancellations [Eq. (24), Fig. 8] when considering the double time-branch. Likewise, the nontrivial CP constraint on Keldysh diagrams is achieved by grouping their two-branch diagrams according to one-branch time-ordering [Eq. (21), Fig. 7(b)].

*Approximations.* It is no surprise that our simple rules in general lead to quantities that are difficult to calculate. However, the merit of the approach is that it indicates new routes to systematically address the open problems of devising and comparing CP approximations, improving positivity in existing TP approximations, and further exploring the most challenging issue of CP-TP approximations [43, 44, 62, 63]. The very recent advances reported in Ref. [62, 63] indicate that there is still room for further progress which may benefit from statistical field-theoretical techniques.

Finally, we point out some caveats that may inhibit a useful exchange of ideas between the quantum information and statistical field theory communities. These are also related to the CP-TP duality and its implications for practical approximation schemes. In particular, we have seen that quantum information's insistence on operational clarity in CP approximations comes at the price of great complexity which restricts the sophistication of practical calculations. This makes it tempting to take the 'easy way out' and restrict the physical scope to open-system models that are either exactly solvable or reducible to GKSL limits. As a result, the accuracy of these approximations may in the end be worse than those of well-developed TP approximations in statistical field theory. Yet, a converse critique is also heard: TP approximations take a different 'easy way out' by giving up the operational clarity that is, for example, necessary to correctly account for entanglement in open-system dynamics.

To be sure, both these strategies are valid and deserve to be pursued depending on the specific problem at hand and the concrete research interests. However, we emphasize that ultimately *both* the CP and TP property are desired and should be accordingly monitored. In CP approximations, the TP error does not need to be estimated because it corresponds to the probabilities of neglected physical / operational processes. In contrast, the CP errors in TP approximations are less straightforwardly monitored because they may partially cancel. Nevertheless, the CP property can at least be checked a *posteriori* from the Choi-Jamiołkowski operator, but none of the works employing statistical field theory that we consulted –including earlier work of our own– do this. Importantly, even if this test fails, the framework presented in this paper provides an explicit diagrammatic means to improve upon CP which was so far lacking. Likewise, the quantum information community may benefit from our work as it further paves the way for operationally sound approximations that broaden the scope beyond exactly solvable dynamics or GKSL limits.

## Acknowledgements

We acknowledge B. Criger, D. P. DiVincenzo, M. Pletyukhov, R. Saptsov, H. Schoeller and N. Schuch for useful discussions and K. Nestmann for proofreading the manuscript. We furthermore thank R. van Leeuwen for discussion of Ref. [58] and hospitality extended at the University of Jyväskylä.

**Funding information**  V. R. was supported by the Deutsche Forschungsgemeinschaft (RTG 1995).

## A  Normal-ordering

The normal-ordering $:\ :$ of a string of field operators $X_1 \ldots X_q$ is defined relative to some average $\langle\ \rangle$. In the main text we mostly consider $\langle\bullet\rangle = \mathrm{Tr}_{\mathrm{E}}\,\bullet\,\rho^{\mathrm{E}}$ where $\rho^{\mathrm{E}}$ is a grand canonical equilibrium state of noninteracting fermions or bosons. In the following we focus on this case as well, but note that partial contractions and normal-ordering can be generalized to interacting environments [49, 81].

The construction of normal-ordered expressions follows an iterative reduction into pair-contractions $\overline{X_1 X_2} \equiv \langle X_1 X_2\rangle$. This is based on the observation that the total average $\langle X_1 \ldots X_q\rangle$ decomposes into pair contractions by Wick's theorem. With the assumption $\langle X_i\rangle = 0$, one recursively defines

$$X_1 = :X_1: \tag{54a}$$

$$X_1 X_2 = :X_1 X_2: + \overline{X_1 X_2} \tag{54b}$$

$$X_1 X_2 X_3 = :X_1 X_2 X_3: + \overline{X_1 X_2}\,:X_3: + \overline{X_1\,:X_2:X_3} + :X_1:\overline{X_2 X_3} \tag{54c}$$

$$\vdots$$

and inductively verifies that the average of $:X_1 \ldots X_q:$ is zero for each $q$. Moreover, each *partial* average is zero, for example, $:\overline{X_1 X_2 X_3}: = 0$. For this to work, the partial contractions must include a sign for disentangling in the case of fermions, $\overline{X_1\,:X_2:X_3} \equiv -\overline{X_1 X_3}\,:X_2:$. Note that for $\langle X_i\rangle \neq 0$ the recursive construction must be extended by substituting $X_i \to X_i - \langle X_i\rangle\mathbb{1}$ in Eq. (54) together with $:\mathbb{1}: = 0$.

The sole purpose of normal-ordering is to reorganize the way contractions of $X_1 \ldots X_q$ are performed: operators within a normal-ordered sequence can only be contracted with operators from a different sequence. Thus, $\langle :X_1 \ldots X_q: :Y_1 \ldots Y_q: \rangle$ is evaluated by Wick's theorem in the usual way except that the normal-ordering restricts all pair contractions to be of the type $\langle X_i Y_j\rangle$. The key point used in Eq. (12) is that such an average is nonzero only when the number of operators and the modes on which they act match. Moreover, such an average does *not* require the calculation of the operators $:X_1 \ldots X_q:$ and $:Y_1 \ldots Y_q:$.

In the main text and in App. B we use this property of normal-ordering to decompose any operator $A_m$ acting only on a definite set of environment modes $m = m_q \ldots m_1$ into its environment average plus all its *normal-ordered* partial contractions which systematically account for all possible fluctuations:

$$A_m = \langle A_m\rangle + :\Big(A_m + \sum_{l \subset m} A_m^l\Big): \quad . \tag{55}$$

Here the shorthand notation $l \subset m$ denotes a subset of modes on which the remaining uncontracted fields in $A_m^l$ act and we sum over all possible subsets excluding the empty set $l = 0$ (not indicated).

# B  Keldysh operators

## B.1  Decomposition of unitary into average plus fluctuations

Here, we explain the construction of the Keldysh operators $k_m(t)$ in Eq. (19) from the unitary evolution operators $U(t, t') = T \exp[-i \int_{t'}^{t} d\tau H^{\text{tot}}(\tau)]$ governed by the total Hamiltonian $H^{\text{tot}}(t) = H^0(t) + V(t)$. The uncoupled system plus environment Hamiltonian $H^0$ generates their free evolution $u(t, t') = T \exp[-i \int_{t'}^{t} d\tau H^0(\tau)]$. The coupling is assumed to be normal-ordered, $V = :V:$ and can be expanded as $V(t) = \sum_{m \neq 0} V_m(t)$ into normal-ordered terms $V_m = :V_m:$ acting only on environment modes $m$ since $V_m^l = 0$ and $V_0 = \langle V \rangle = 0$ in Eq. (55). We first expand the unitary in powers of the couplings $V$ and then expand each coupling into its mode contributions,

$$
\begin{aligned}
U &= u + u * (-iV)u + u * (-iV)u * (-iV)u + \dots \\
&= u + \sum_{\mu \neq 0} u * (-iV_\mu)u + \sum_{\mu' \mu \neq 0} u * (-iV_{\mu'})u * (-iV_\mu)u + \dots \\
&= U_0 + \sum_{m_1} U_{m_1} + \sum_{m_2 m_1} U_{m_2 m_1} + \dots = \sum_m U_m,
\end{aligned}
$$ 
(56)

writing $[u * Vu](t, t') = \int_{t'}^{t} d\tau u(t, \tau) V(\tau) u(\tau, t')$. In the last line we then collect into $U_m$ all strings of repeated couplings which together act on the definite set of[30] environment modes $m = m_q \dots m_1$. We do not distinguish the order of modes, i.e., $U_{m_q \dots m_1}$ is symmmetric. Next, we *decompose*[31] $U$ by expanding each component

$$
U_m = \langle U_m \rangle + :\left( U_m + \sum_{m' \subset m} U_m^{m'} \right):
$$ 
(57)

into its average plus fluctuations [Eq. (55)]. Inserting into Eq. (56), we collect all terms with the fixed set of modes $m$ originating from supersets $m' \supset m$ in the expansion (57) and obtain

$$
U = \sum_m U_m \equiv k_0 + \sum_{m \neq 0} :k_m: \quad .
$$ 
(58)

Here, the Keldysh operator $k_m$ is defined for $m \neq 0$ as the partially contracted evolution operator $U$ such that only environment modes $m \subset m'$ remain:

$$
k_m \equiv U_m + \sum_{m' \supset m} U_{m'}^m.
$$ 
(59)

The special case denoted by $m = 0$ is the part of $U$ in which no environment operators are left: it collects the full contraction of $U$, i.e., its *environment average*

$$
k_0 \equiv \sum_m \langle U_m \rangle = \langle U \rangle.
$$ 
(60)

We note that the Keldysh operators are themselves not normal-ordered: $k_m \neq :k_m:$, i.e., $k_m$ has nonzero partial contractions for $m \neq 0$ while $k_0 = \langle k_0 \rangle \otimes \mathbb{1}_{\text{E}}$ and $:k_0: = 0$. Yet, partial contractions of these objects never appear in the theory because the pasting rule Eq. (21b) explicitly normal-orders the Keldysh operators.

---

[30]The discussion applies to any normal-ordered coupling $V$. Only for the special case of bilinear coupling, does the number of modes count the coupling power, $U_{m_1} = u * V_{m_1} u$, $U_{m_2 m_1} = u * V_{m_2} u * V_{m_1} u$, and so on. If $V$ is instead bi-quadratic as, for example, in Appelbaum-Schrieffer-Wolf Hamiltonians describing Kondo physics, then $U_{m_2 m_1}$ may be of first order in the coupling $V$.

[31]We are *not* normal-ordering $U$, i.e., *altering* $U \to :U:$. In Eq. (58) we merely add and subtract canceling partial contractions in the real-time expansion of $U$ to separate out different kinds of fluctuations.

## B.2 Consistency with the Schrödinger equation

The main text noted that summing the hierarchy (32) results in a self-consistent equation,

$$k \equiv k_0 + k_1 + k_2 + \ldots = k_0 + k_0 \sum_{l=1}^{\infty} [*(\sigma_1 + \sigma_2 + \ldots) * k_0]^l \tag{61a}$$

$$= u + u \sum_{l=0}^{\infty} [*(\sigma_0 + \sigma_1 + \sigma_2 + \ldots) * u]^l = u + u * \sigma * k, \tag{61b}$$

with the sum of all self-energies $\sigma \equiv \sum_m \sigma_m$. Importantly, this is *not* the self-consistent form of Schrödinger equation because of the lack of explicit normal-ordering, $k(t) \neq U(t)$ and also $\sigma(t, t') \neq -iV(t)\delta(t - t')$.

To check that the hierarchy (32), when properly normal-ordered, indeed unravels the Schrödinger equation, we must decompose $k = k_0 + (k - k_0)$ and separately consider the operator $k - k_0$ acting nontrivially on the environment. From Eq. (61a) we find that $k$ also obeys the self-consistent equation

$$k = k_0 + k_0 * (\sigma - \sigma_0) * k. \tag{62}$$

Subtracting $k_0$, we rewrite the right hand side by inserting the zeroth hierarchy equation (32a), $k_0 = u + u * \sigma_0 * k_0$, and again substitute Eq. (62) in the resulting term:

$$\begin{aligned} k - k_0 &= k_0 * (\sigma - \sigma_0) * k \\ &= u * (\sigma - \sigma_0) * k + u * \sigma_0 * k_0 * (\sigma - \sigma_0) * k \\ &= u * (\sigma - \sigma_0) * k + u * \sigma_0 * (k - k_0) = u * (\sigma * k - \sigma_0 * k_0). \end{aligned} \tag{63}$$

From this one can reproduce the self-consistent form of Schrödinger equation as follows. Starting from its right hand side, insert $U = k_0 + :k - k_0:$ [cf. Eq. (58)]. Next, decompose the result into its average plus normal-ordered expressions:

$$\begin{aligned} u + u * (-iV)U &= u + u * (-iV)(k_0 + :k - k_0:) \\ &= u + u * \sigma_0 * k_0 + u * :\sigma * k: \end{aligned} \tag{64a}$$

$$= k_0 + :k - k_0: = k_0 + \sum_{m \neq 0} :k_m: = U. \tag{64b}$$

As a result of the hierarchy equations this is consistent with the decomposition (58): to obtain Eq. (64b) we used $k_0 = u + u * \sigma_0 * k_0$ again and inserted the *normal-ordering* of Eq. (63), noting that $:\sigma_0 * k_0: = 0$. Thus, $k_0$ together with the *normal-ordering* of $\sum_{m \neq 0} k_m$ as determined by the hierarchy (32) are equivalent to the self-consistent form of the Schrödinger equation.

The step leading to Eq. (64a) involves a nontrivial reorganization of terms,

$$(-iV)(k_0 + :k - k_0:) = \sum_{n \neq 0} (-iV_n)\Big(k_0 + \sum_{m \neq 0} :k_m:\Big) \tag{65a}$$

$$= \sum_{n \neq 0} \langle -iV_n :k_n: \rangle + :\Big[ \sum_m \sum_{l \subseteq nm} (-iV_n :k_m:)^l \Big]: \tag{65b}$$

$$= \sigma_0 * k_0 + :\Big[ \sum_{n'm'} \sigma_{n'm'} * \sum_{m''} k_{m''} \Big]: = \sigma_0 * k_0 + :\sigma * k:, \tag{65c}$$

which involves the following detailed diagrammatic observations.

First, the expansion (55) is applied to Eq. (65a). The first term in Eq. (65b) is the average which sums up all full contractions of fields in $V_n = :V_n:$ with those in $:k_n:$. The corresponding diagrams can be factorized into two parts. One factor is irreducibly contracted in some

time interval $[t, t']$ such that summing all contributions to it gives the self-energy component $\sigma_0(t, t')$. The remaining factor for the time interval $[t', 0]$ contains all possible contributions without external modes and therefore sums up to $k_0(t')$. Since in $k_n(t)$ we integrate over all internal times, we also integrate over $t'$ and obtain the convolution in the first term in Eq. (65c).

The second term in Eq. (65b) involves[32] $(-iV_n : k_m :)^l$ where a subset $l \subseteq nm$ of modes in $V_n$ and $:k_m:$ remains uncontracted. Denote by $c$ the –possibly empty[33] – subset $c \subseteq n$ of modes in $V_n$ that are contracted with a corresponding subset $c \subseteq m$ of modes in $k_m$. Again the diagrams factorize. One factor contains all diagrams *irreducibly* connected to $V_n$ in some time-interval $[t, t']$ leaving modes $n'$ from $V_n$ and modes $m'$ from $k_m$ uncontracted. Summing over all modes $c$, we get by definition $\sigma_{n'm'}(t, t')$ representing all such irreducible contributions. The factor at earlier times $[t', 0]$ contains all possible contributions with further uncontracted modes $m''$ from $k_m$ and therefore sums up to $k_{m''}(t')$.

Thus, we have decomposed the modes in $V_n$ as $n = n'c$ and in $k_m$ as $m = cm'm''$ such that the uncontracted modes $n'm'$ and $m''$ are attributed to the irreducible and reducible factors of each diagram, respectively. In total, the modes $l = n'm'm''$ thereby remain uncontracted. Independently summing over the modes $n'm'$ and $m''$ –including the empty sets– we obtain $\sigma * k$ [Eq. (61)] for the second term in Eq. (65c), completing the derivation.

## C   CP-TP duality: State-evolution correspondence

The duality between CP and TP constraints noted in Ref. [70] can be understood in a general form by combining insights from quantum-information theory and statistical field theory. In the main text we focused on pin-pointing the *microscopic* origin of this CP-TP duality, i.e., on the level of individual diagrammatic contributions to the Keldysh expansion, rather than using the formal arguments that we now summarize.

In general, the state-evolution connection is provided by the de Pillis-Jamiołkowski-Choi map (7) which turns a *superoperator* $\Pi$ into an *operator* on the double system:

$$\mathrm{choi}(\Pi) = (\Pi \otimes \mathcal{I})|\mathbb{1}\rangle\langle\mathbb{1}|. \tag{66}$$

Here $|\mathbb{1}\rangle = \sum_k |k\rangle \otimes |k\rangle$ is the maximally entangled state on the double system. Thus *evolutions* are mapped to *states*, providing many helpful insights [73].

*Kraus form.* When the propagator is written in Kraus form, $\Pi = \sum_m K_m \bullet K_m^\dagger$, the Choi operator reads $\mathrm{choi}(\Pi) = \sum_m |K_m\rangle\langle K_m|$ with vectorizations of the Kraus operators $|K_m\rangle := (K_m \otimes \mathbb{1})|\mathbb{1}\rangle$. One independently shows that $\Pi$ is a CP evolution superoperator if and only if $\mathrm{choi}(\Pi)$ is a positive (semidefinite) operator and thus represents some quantum state of the double system. The Kraus form of $\Pi$ translates CP into a property which can be assessed by inspecting individual terms: physically, it is clear that the statistical mixing of individual pure states $|K_m\rangle\langle K_m|$, always leads to a positive state $\mathrm{choi}(\Pi)$.

On the other hand, one verifies from Eq. (66) that $\Pi$ is a TP superoperator if and only if the partial trace of the operator $\mathrm{choi}(\Pi)$ over the first subspace of the tensor product gives the identity on the second:

$$\underset{1}{\mathrm{Tr}}\,\mathrm{choi}(\Pi) = \mathbb{1}. \tag{67}$$

Physically speaking, this requires that the Choi state (which is already a mixed state) has the maximally mixed state as a marginal state, i.e., Eq. (67) constrains the *correlations* on the

---

[32]The first term $(-iV)k_0$ in Eq. (65a) is included through the term $m = 0$ of Eq. (65b).

[33]For $c = 0$ (no contractions) we have $n' = n$ and $m'' = m$ and $m' = 0$ and obtain $-iV_n\delta$ as a contribution to the self-energy $\sigma_{n'm'} = \sigma_n$, convoluted with the reducible part $k_{m''} = k_m$.

bipartite system. This is clearly a nontrivial property to enforce on a decomposition into pure states: indeed, inserting the Kraus form one obtains $\sum_m \mathrm{Tr}_1\,|K_m\rangle\langle K_m| = \mathbb{1}$ which is equivalent to the condition $\sum_m K_m^\dagger K_m = \mathbb{1}$ mentioned in the main text [Eq. (17a)] and puts a constraint on *all* terms.

*Keldysh Dyson expansion.* Now consider the same superoperator $\Pi$ written in the form of a Dyson series $\Pi = \pi + \sum_{k=1}^{\infty}(\pi * \Sigma *)^k \pi$ with respect to the physical free evolution $\pi$ which is CP-TP. The TP property of $\pi$ guarantees that $\Pi$ is TP as well, provided that the remaining terms have zero trace, $\mathrm{Tr}\,\Sigma = 0$. This condition is guaranteed by simple pairwise cancellation of diagrams discussed in Sec. 3.5.

However, the CP constraint is nontrivial due to the complicated form $\mathrm{choi}(\Pi) = \mathrm{choi}(\pi) + \sum_{k=1}^{\infty} \mathrm{choi}\big((\pi * \Sigma *)^k \pi\big)$ of the Choi operator inherited from $\Pi$: although $\mathrm{choi}(\pi)$ is positive due to the complete positivity of $\pi$, the remaining terms do not make clear that $\mathrm{choi}(\Pi)$ is indeed positive and, consequently, that $\Pi$ is CP.

We thus see that the state-evolution correspondence (66) changes the difficulty of ensuring CP *at the expense* of ensuring TP and *vice-versa*. This explains the noted [70] 'incompatibility' of simultaneously imposing these constraints in practice.

# D  Diagrammatic rules for conditional evolution $\Pi_m$

In the main text we focused on the reorganization of the Keldysh expansion [Eq. (21)]. The explicit form of the terms becomes most clear in the interaction picture:

$$\Pi(t) = \mathrm{Tr}_{\mathrm{E}} \sum_n \int_{\tau_n \geq \dots \geq \tau_1} [-iV(\tau_n)]\dots[-iV(\tau_1)] \bullet \otimes \rho^{\mathrm{E}} \sum_{n'} \int_{\tau'_{n'} \geq \dots \geq \tau'_1} [iV(\tau'_1)]\dots[iV(\tau'_{n'})] \quad (68)$$

$$= \sum_k \sum_{\mathrm{diagrams}} \int_{t_k \geq \dots \geq t_1} \mathrm{Tr}_{\mathrm{E}} \dots [-iV(t_i)]\dots \bullet \otimes \rho^{\mathrm{E}} \dots [iV(t_j)]\dots. \quad (69)$$

The diagrammatic rules for the contributions to Eq. (69) are the following:

(a) Distribute $n$ [$n'$] vertices on the left (right) in all possible ways over the $k = n + n'$ ordered times $t_k \geq \dots \geq t_1$ and sum over these diagrams.

(b) Pairwise contract all fields appearing in the vertices $V$ and sum over all possible contractions. The sum over all modes is included in the vertices themselves, $V = \sum_{m \neq 0} V_m$. For fermions, the sign of each term is given by the parity $(-1)^{n_{\mathrm{c}}}$ of the number $n_{\mathrm{c}}$ of crossing contraction lines.

(c) Perform the ordered time integrations.

(d) To obtain the conditional propagator $\Pi_{m_q \dots m_1}$, restrict in (b) the $q$ inter-branch contractions to a definite set of modes $m_q \dots m_1$ while summing over all modes of intra-branch contractions. For $\Sigma_{m_q \dots m_1}$ restrict this to two-contour irreducible diagrams.

Remarks:

(i) The times $t_i$ label vertices on *either* contour: in the Keldysh technique each *two-branch time-ordered* integral is explicitly represented by a separate diagram. This time-ordering allows the transition to a Laplace frequency conjugate to the *physical time* in $\rho(t)$ which is of practical use in calculations [19, 20, 22, 23, 26, 28] and crucial for the formulation of RG schemes [34–36, 39–41]. The conversion between single-contour time-ordering ($\tau_i$) used in Eq. (68) to two-contour time-ordering ($t_i$) in Eq. (69) as is discussed in the main text [Sec. 4.3] has also been useful in other contexts [23].

(ii) In the density-*operator* technique the Keldysh contour is not drawn closed at latest time [cf. Fig. 5(b)], in contrast to the Keldysh contour for Green's functions [58] where this closure

indicates an additional trace over the system to obtain correlation *functions*.

(iii) For the environments of interest here, the modes are labeled by discrete indices (particle type, electron spin, etc.) and a continuous orbital index or equivalent energy [34, 41], see Eq. (6). Thus in the above rules, the CP structure requires *suspending integrations* over the continuous environment energies for the inter-branch contractions but performing them for intra-branch contractions, see Sec. 5.

In practice, to compute the superoperators represented by the diagrams one needs to exploit the known energy eigenbases of $H$ and $H^{\mathrm{E}}$, the system and environment Hamiltonians, respectively. It is then convenient to revert to the Schrödinger picture which only changes the translation rules for the diagrams by inserting the free evolution

$$\pi(t, t') = T e^{-i \int_{t'}^{t} d\tau H(\tau)} \bullet T' e^{+i \int_{t'}^{t} d\tau H(\tau)}, \tag{70}$$

with formal (anti)time-ordering $T$ ($T'$) between every two consecutive occurences of vertices at times $t \geq t'$ –irrespective of the branch they occur on.

# E  Diagrammatic rules for Keldysh operators $k_m$

A key result of the paper [Eq. (21b)] is that the conditional propagator $\Pi_{m_q \ldots m_1}$, determined by the rules in App. D, can be 'cut into two halves', representing the *Keldysh operator* $k_{m_q \ldots m_1}$ and its adjoint, respectively. The Keldysh operators can therefore be obtained directly from a diagrammatic 'cutting and pasting' based on similar rules.

## E.1  Cutting rules

The contributions to the Keldysh operator $k_{m_q \ldots m_1}(t)$ are obtained from the formal expansion of the auxiliary operator

$$\sum_n \mathrm{Tr}_{\mathrm{E}} \int_{\tau_n \geq \ldots \geq \tau_1} d\tau_n \ldots d\tau_1 [-iV(\tau_n)] \ldots [-iV(\tau_1)] \rho^{\mathrm{E}}, \tag{71}$$

using the following rules:

(a) Pairwise contract all fields appearing in the vertices $V$ and sum over all possible contractions, leaving $q$ fields with a definite set of mode indices $m_q \ldots m_1$ *uncontracted*. For fermions, the number of crossing contraction lines $n_{\mathrm{c,intra}}$ introduces a sign $(-1)^{n_{\mathrm{c,intra}}}$, where the crossings with the external contraction lines must be *included*.

(b) Assign the ordered times $\tau_q \geq \ldots \geq \tau_1$ to the uncontracted vertices and integrate over them.

(c) Independently sum over all orderings of the mode indices $m_q \ldots m_1$.

To obtain the self-energies $\sigma_{m_q \ldots m_1}$, restrict to *one-branch irreducible* diagrams.

The central objects are the Keldysh operators $k_m$ and their self-energies $\sigma_m$. The Kraus operators can be obtained from these,

$$K^e_{m_q \ldots m_1} = \langle e | : k_{m_q \ldots m_1} : \otimes \mathbb{1}_{\mathrm{E}'} | 0 \rangle = \mathrm{Tr}_E \, \hat{e} : k_{m_q \ldots m_1} : \sqrt{\rho^{\mathrm{E}}},$$

where $|e\rangle = \sum_{ij} e_{ij} |i\rangle \otimes |j\rangle$ denotes any desired basis of bipartite vectors for the purified environment EE' and $\hat{e} = \sum_{ij} e_{ij} |i\rangle \langle j|$ is a corresponding basis of operators acting on E.

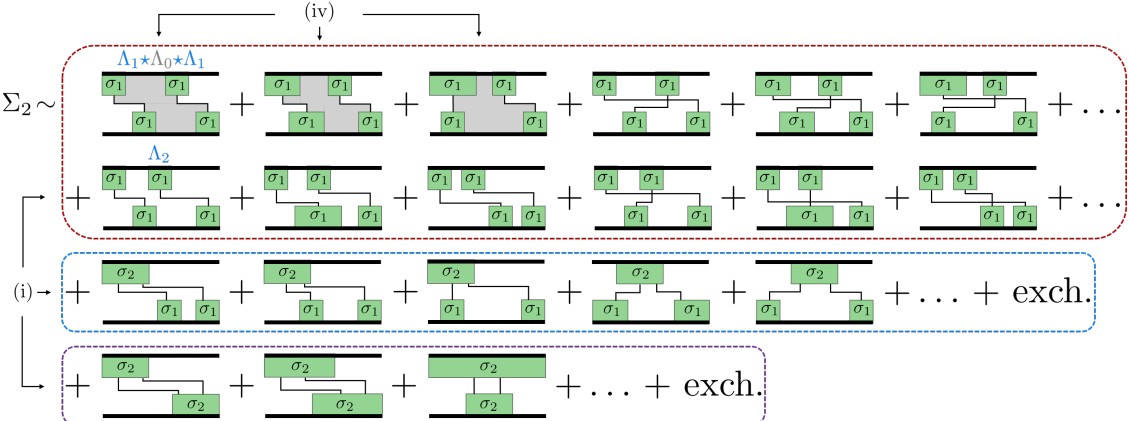

Figure 13: Contributions to $\Sigma_2$ based on the diagrammatic rules (i)-(v) in Sec. 4.3. The bold lines indicate a renormalized propagation $k_0$ $[k_0^\dagger]$ on the upper (lower) branch. Each diagram represents a *definite* two-branch time-ordering, i.e. the time-integrations over the internal times $\tau_+, \tau'_+$ $[\tau_-, \tau'_-]$ on the upper [lower] branch are restricted as to not overtake each other. The colored boxes group the different distributions of the external vertices over generators $\sigma_\mu$ by rule (i). The first three diagrams are build up from reducible diagrams which are forced to be irreducible by inserting fragments $\Lambda_0$ (gray) by rule (iv).

## E.2 Pasting rules

The Keldysh real-time diagrams for $\Pi_{m_q\ldots m_1}$ can be obtained from the above constructed Keldysh operators $k_{m_q\ldots m_1}$ by 'pasting' them together with the vertically flipped diagrams corresponding to $k_{m_q\ldots m_1}^\dagger$. This diagrammatic pasting entails the following rules:

(a) Contract in all possible ways the external vertices of $k_{m_q\ldots m_1}$ with those of $k_{m_q\ldots m_1}^\dagger$ having the same mode index, remembering that $m_q\ldots m_1$ may contain repetitions of the same mode index. For fermions, the number of crossing inter-branch contraction lines $n_{c,inter}$ introduces an additional sign $(-1)^{n_{c,inter}}$. This produces the correct fermionic sign in App. D since $n_c = n_{c,intra} + n_{c,inter}$.

(b) Assign the ordered times $\tau_{2q} \geq \ldots \geq \tau_1$ to the $2q$ consecutive external vertices – irrespective of the branch they occur on– and integrate over them. This produces the two-branch time-ordering in App. D and yields $\Pi_{m_q\ldots m_1}$.

(c) Sum over all modes $m_q\ldots m_1$ of the inter-branch contractions to obtain $\Pi$.

## F  Functional relation of self-energies $\Sigma_m$ and $\sigma_m$

The general rules (i)-(v) of Sec. 4.3 for constructing the conditional self-energies $\Sigma_m$ from the memory-kernels $\sigma_m$ are illustrated using Fig. 13. As pointed out in the main text, the functional form (39a) for $\Sigma_0$ is the only contribution requiring an *infinite* set of diagrams to be summed up. All following tiers require only a *finite* set of dressed diagrams constructed from $\Sigma_0$. Here we explicitly discuss the next-to-leading order $\Sigma_2$ which covers all difficulties to be encountered when applying the rules to these higher orders.

(i) We first distribute in all possible ways the external vertices among generators $\sigma_\mu$ with $\mu \neq 0$ on each branch. In Fig. 13, the colored boxes group the three possible distributions for $\Sigma_2$: The first group (red) has two $\sigma_1$ blocks on each branch. The second group (blue) has two $\sigma_1$ blocks on one branch but a single $\sigma_2$ on the opposite branch. The final possibility (violet)

has a single $\sigma_2$ on each branch.

(ii) Within each of these groups, we sum over diagrams with *definite two-branch time-ordering* and integrate over all internal time-arguments. We stress that the number of such orderings is *finite*, although we refrain from providing all contributions for $\Sigma_2$.

(iii) We contract the individual generators $\sigma_\mu$ in all possible ways between the branches, including 'exchange' diagrams that are explicitly shown only in the first (red) group.

(iv) To proceed, we have to distinguish two types of diagrams that we have constructed so far. Two-branch *reducible* contributions [first three diagrams in Fig. 13] are constructed from multiple *two-branch irreducible fragments* $\Lambda_\mu(t_+, t_-, t'_+, t'_-)$ 'glued' together to enforce the two-branch irreducibility. This is achieved using the four-time generalization, $\Lambda_0(t_+, t_-, t'_+, t'_-)$, of the self-energy $\Sigma_0(t, t')$ [cf. Eq. (39a)] indicated in gray. Together with the two-branch convolution $\star$, we obtain terms of the form

$$\Sigma_2(t, t') \sim [\Lambda_1 \star \Lambda_0 \star \Lambda_1](t, t, t', t') \tag{72}$$

for these contributions. The remaining *irreducible* contributions shown in Fig. 13 are made up of already irreducible fragments and require no such 'fixing':

$$\Sigma_2(t, t') \sim \Lambda_2(t, t, t', t'). \tag{73}$$

(v) Finally, all constructed diagrams have to be complemented by their possible extensions with $\Lambda_0$ blocks on either side or both, as discussed in the main text [Eq. (39)]. These are not shown in Fig. 13.

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
