# Peer review of "Density-operator evolution: Complete positivity and the Keldysh real-time expansion"

_SciPost Physics, doi:SciPost Phys. 7, 012 (2019)_

## Round 1 · Referee Report · Anonymous (Referee 1) · 2019-5-11

Report

The manuscript contains a thorough formal analysis of the density-operator evolution of a quantum system coupled to an environment by comparing quantum-information and statistical field-theory approaches. On a very general level the authors discuss the conditions for complete positivity (CP) and trace-preservation (TP) to be guaranteed and propose approximation schemes how to fulfil these two important properties. This is a very challenging issue and of high importance. Although the authors discuss in the end only a very trivial and exactly solvable method, there is a high need to understand from a formal point of view which approximation schemes fulfil CP and/or TP, what is missing in a certain approximation, and up to what extent CP and TP are fulfilled. Although in quantum-information based approaches it is well-known how to fulfil CP and TP by writing the time evolution in terms of Kraus operators which have to fulfil a certain sum rule, the set of models which can be solved explicitly in this way is very limited and usually reduces to almost trivial and exactly solvable cases. Therefore, the goal of the present manuscript to connect quantum-information based approaches to real-time diagrammatic techniques is very important since only the latter have proven to be promising tools to find solutions of nontrivial models and to go significantly beyond phenomenological ansatzes (like, e.g., Lindblad master equations).

The new important insight of the manuscript is the reorganization of Keldysh diagrams in terms of Kraus operators by distinguishing carefully between contractions within a single branch of the Keldysh contour and diagrams connecting the two branches, outlined in Section 3. Only the latter are shown to correpond to physical processes in terms of measurements of the environment, whereas the former are important to define the Kraus operators. This leads to the central equation (17). In addition, a whole set of hierachical equations is proposed in Section 4 of how to calculate the Kraus operators in terms of single-branch irreducible diagrams and how to relate the full propagator to two-branch irreducible self-energies (defining the memory kernel of the kinetic equation when summed over all processes). Although the explicit solution of these equations is certainly very challenging, it is a very important first step to state them clearly and to analyse precisely what is needed to fulfil TP and CP. This might not only be important for analytical calculations (e.g., via perturbative approaches) but also for purely numerical approaches, where the reorganization of terms with the help of quantum-information based insights has proven to be very crucial and successful in tensor network based methods.

The manuscript is written very carefully and presents in addition a useful and complete overview over many other methods used in the literature to deal with open quantum systems. In conclusion, I strongly recommend publication of the present manuscript in SciPost.

Besides this overall positive recommendation, I would like to mention a couple of suggestions which the authors might consider to improve their manuscript:

(1) It might be helpful for the general readership if the authors invest some more space in the beginning to state more clearly the model under consideration and the form of the system-environment coupling. E.g., it is not at all clear what the authors mean precisely by environment modes which are crucial for the whole formal considerations and occur as summation variables in all formulas. A few examples of models at the beginning might be very useful in this respect. E.g., the authors mention at some points in the manuscript "bilinear" and "biquadratic" couplings. As far as I understand "bilinear" means that the coupling involves one field operator of the environment and one from the quantum system. However, such couplings are also called "linear" in other works (w.r.t. the environment). Of course the authors are free to choose whatever they prefer the coupling to be called but it should be clear at least what they mean.

(2) In the introduction and throughout the manuscript the authors emphasize several times that certain ways to deal with open quantum systems can be divided in "physical" and "mathematical" approaches. It may be a matter of taste to present it in this form but in my opinion this strict classification is rather dangerous. The authors give the readership the impression that CP has highest priority and that all other approaches which do not guarantee CP are unphysical, meaningless and purely mathematical. This is certainly not true and depends crucially on the definition of "physical" and "unphysical" being a very vague and community-dependent issue. The authors adapt here the point of view of the quantum-information based community but I am sure that many researchers from the field-theoretical side might have another opinion. E.g., the irreducible memory kernel defining the kinetic equation has a clear physical meaning, it is a retarded response function relating the density matrix at some former time to the time derivative of the density matrix at the time of observation. As shown in many previous works its Fourier transform has the corresponding analytic properties of a retarded response function and its branching points give directly the Rabi frequencies and all dissipative decay rates of the system. It is absolutely unclear how such simple relations can be extracted from the Kraus operators. Moreover, the authors themselves discuss in detail the CP-TP duality and state very clearly that approximations fulfilling CP often violate TP, which one can also call very "unphysical" (i.e., normalization of probabilites is not guaranteed). From my point of view, one should soften some statements in the manuscript in this regard. It is a well-known fact that all kinds of sum rules are nice to have and it is important to understand in which approximation scheme they are valid and in which one not. But often such formal considerations lead to hardly solvable self-consistent sets of equations and it is not guaranteed at all that the resulting solution is close to the correct result although fulfilling all sum rules. Therefore, it is often much better to disregard certain sum rules and to set up approximations in terms of systematic expansion parameters such that all sum rules are fulfilled up to a small and controllable error.

(3) In Section 3.4 the authors discuss how TP can be fulfilled by summing over the two possibilities that the last vertex lies on the upper or lower branch of the Keldysh contour. This shows very clearly that TP is broken within naive CP-approximation schemes by just restricting the calculation of the propagator to a certain subset of "physical" processes (see also (2) above). In addition, I would like to mention that it is known that similiar cancellations can also occur by changing the position of several vertices between the two branches, e.g., for noninteracting systems. This shows that schemes fulfilling just CP are very dangerous and can lead to completely wrong solutions even for almost trivial models (e.g. Kondo physics can easily be generated in noninteracting systems, a fact very well-known in quantum impurity problems). The authors might consider to mention this point.

(4) After Eq.(21) the authors mention that setting a cutoff for the number of contractions connecting the two Keldysh branches does not correspond to a naive short-time expansion of the propagator. I was not able to understand their statement what kind of influence this cutoff has on the quality of the time evolution for all times. The readers might appreciate a more precise statement, instead of just saying that the simple example in Section 5 will illustrate it.
  • validity: -
  • significance: -
  • originality: -
  • clarity: -
  • formatting: -
  • grammar: -

Author:  Maarten Wegewijs  on 2019-06-06  [id 533]

(in reply to Report 1 on 2019-05-11)

We thank the referee for carefully reading our manuscript, as reflected by the pertinent summary. The suggestions provided by the referee were very helpful in further improving the manuscript, specifically in both balancing and sharpening the conclusions. We have provided a separate PDF of the significantly revised manuscript which marks all changes in blue.

Answer to (1):

In the revised manuscript we included in the introduction a new section “Microscopic models for open-system dynamics” which addresses these points.

Answer to (2):

We admit that in the original manuscript we deliberately polarized the discussion to make the following point: Although both approaches/communities have their own merits, they also have their own problems. Also, their mutual points of critique are valid and neither community has at present an all-encompassing solution for the identified problems. To avoid any impression of a bias, we explicitly indicate in the revised manuscript whenever we polarize/simplify for the sake of clarity.

We have also significantly revised the Introduction, Sec. 2 and the Summary to further balance the discussion. We have adjusted the phrasings mentioned by the referee to avoid unnecessary debates. The added discussion in the Summary tries to make clearer that a comparison of the two approaches involves a trade-off that is not so simple, and we address some issues concerning vague and community-dependent definitions pointed out by the referee.

For sure, we do not wish to convey that either the complete positivity (CP) or trace-preservation (TP) property has a higher priority in general. We also do not want to leave the impression that approximations that fail either CP or TP are useless. A message that we do want to convey is that one should start thinking more seriously about CP and its role in accounting for dynamics in the presence of entanglement. As the referee noted, we provide a first crucial step in this direction by finally making quantum information results relevant to field-theoretical methods on the most general level. In the revised manuscript we included a new section “Entanglement, dynamics and complete positivity” at the beginning of the quantum information discussion to more clearly point out the importance of entanglement. Throughout the revised manuscript we still point out serious issues regarding CP that are often overlooked, misunderstood or simply ignored. At the same time, the revised manuscript emphasizes more strongly the equally serious practical advantages of existing TP approximations. Nevertheless, we believe that the encountered difficulties should not discourage one to further pursue the idea of CP approximations.

Answer to (2) "E.g., the irreducible memory kernel defining the kinetic equation has a clear physical meaning,..."

This is a very interesting case in point, and we can follow much of the referee’s argument. In the revised manuscript we now point out the Laplace/Fourier frequency structure of the memory kernel as fundamentally important with appropriate references based on field-theoretical approaches. We also fully agree that the Kraus operators are unsuitable for studying this physical (in the sense of “causal”) structure and we have included this statement. Our results however reveal for the first time the additional structure of the memory kernel which is algebraic and ensures the CP property. This physical (in the sense of “operational”) structure is equally important but unclear from the frequency representation of the memory kernel. This is also mentioned clearly in the revised manuscript.

Answer to (3):

Indeed, TP is broken in CP approximations as we clearly mentioned. To avoid confusion about what this implies we have expanded the “CP vs. TP approximation” section. If we correctly understand the referee, the mentioned additional cancellations are responsible for the exact solvability of our simple model in Sec. 5. We have added a footnote there with a reference that discusses this. In that section we illustrate explicitly how CP and such cancellations interact in a subtle way: The exact solvability is based on exploiting cancellations that tend to hide the CP structure. In the manuscript we already point out that these are “subtle” but see no reason to suggest that these are “dangerous” in some sense. We are aware that the mentioned approximations for Kondo models (using slave-particles) have serious drawbacks and we now mention this in a footnote of the revised manuscript in the introductory section “Approximations”. However, we do not see how they relate to the CP approximations that we discussed here: These in general do not simplify an interacting problem by making it effectively noninteracting as slave particles do. To avoid confusion, the revised manuscript now also mentions this last point in Sec. 1.4.

Answer to (4):

We have expanded and clarified the text after Eq. (21) [Eq. (25) in revised manuscript] and adjusted the discussion in Sec. 5 [after Eq. (53) in revised manuscript] to more clearly support the claim.

Attachment:

revision1_colored_compressed.pdf

---

## Round 2 · Referee Report · Anonymous · 2019-6-12

Report

The authors have undertaken a very careful revision of the manuscript and addressed all my points of concern in a satisfactory and balanced way. It has to be expected that their thorough analysis of CP and TP properties together with the relation to quantum entanglement and expansions in terms of physical processes will bring the two communities working with quantum information based or quantum field theoretical methods closer together. It will sharpen the mind of these two communities w.r.t. the advantages and disadvantages of their methods and motivates the study of further approximation schemes bringing together the various aspects. In this respect I regard the manuscript to be a unique piece of work with a lot of potential of high impact in the field of open quantum systems. Therefore, I strongly recommend publication in SciPost physics.

---

## Round 2 · List of Changes

(PDF with changes marked in color was provided to the editor.)

---

## Editorial Decision

published